# Genomic insights unveil the plasmid transfer mechanism and epidemiology of hypervirulent *Klebsiella pneumoniae* in Vietnam

Quynh Nguyen[1,5], Yen Thi Phuong Nguyen[1,5], Tuyen Thanh Ha[1], Dung Thi Ngoc Tran[1], Phat Vinh Voong[1], Vinh Chau [1], Phuong Luong Nha Nguyen[2], Ngan Thi Quynh Le[2], Lan Phu Huong Nguyen[2], To Thi Nguyen Nguyen[1], Tan Van Trinh[1], Juan J. Carrique-Mas[1,3], Stephen Baker [4], Guy Thwaites [1,3], Maia A. Rabaa[1,3], Marc Choisy[1,3], Hao The Chung [1] & Duy Thanh Pham [1,3] ✉

Hypervirulent *Klebsiella pneumoniae* (hvKp) is a significant cause of severe invasive infections in Vietnam, yet data on its epidemiology, population structure and dynamics are scarce. We screened hvKp isolates from patients with bloodstream infections (BSIs) at a tertiary infectious diseases hospital in Vietnam and healthy individuals, followed by whole genome sequencing and plasmid analysis. Among 700 BSI-causing Kp strains, 100 (14.3%) were hvKp. Thirteen hvKp isolates were identified from 350 rectal swabs of healthy adults; none from 500 rectal swabs of healthy children. The hvKp isolates were genetically diverse, encompassing 17 sequence types (STs), predominantly ST23, ST86 and ST65. Among the 113 hvKp isolates, 14 (12.6%) carried at least one antimicrobial resistance (AMR) gene, largely mediated by IncFII, IncR, and IncA/C plasmids. Notably, the acquisition of AMR conjugative plasmids facilitated horizontal transfer of the non-conjugative virulence plasmid between *K. pneumoniae* strains. Phylogenetic analysis demonstrated hvKp isolates from BSIs and human carriage clustered together, suggesting a significant role of intestinal carriage in hvKp transmission. Enhanced surveillance is crucial to understand the factors driving intestinal carriage and hvKp transmission dynamics for informing preventive measures. Furthermore, we advocate the clinical use of our molecular assay for diagnosing hvKp infections to guide effective management.

*Klebsiella pneumoniae* (Kp) is a Gram-negative pathogen that constitutes a substantial health burden globally. It is estimated that Kp was the second leading bacterial cause of mortality, responsible for more than 200,000 deaths attributed to antimicrobial resistance (AMR) in 2019[1]. There are two distinct pathotypes of Kp: the classical (cKp) and the hypervirulent (hvKp). cKp are a common cause of opportunistic infections in hospital settings, mainly affecting individuals with pre-existing comorbidities. cKp can cause various infections, including

pneumonia, bacteremia, and urinary tract infections, but metastatic dissemination is relatively uncommon[2,3]. Notably, cKp exhibits a wide diversity of K capsular types and a high propensity to develop resistance to commonly used antibiotics, such as cephalosporins, carbapenems, and colistin. This has led to the emergence and worldwide spread of multiple highly resistant lineages (i.e., ST258, ST15, ST11, and ST307)[4–6]. The first report of hvKp was from a patient with pyogenic liver abscesses in Taiwan in 1986[7]. This discovery highlighted a distinctive form of invasive community-acquired Kp infection that can affect healthy individuals without underlying diseases[8]. Beyond liver abscesses, hvKp infections can manifest as endophthalmitis, meningitis, or necrotizing fasciitis, which are typically uncommon in cKp infections[9,10]. Most hvKp infections are caused by K1 (i.e., ST23) and K2 (i.e., ST86 and ST65) capsular serotypes[11,12]. Among these hvKp lineages, ST23/K1 is the predominant lineage responsible for human liver abscesses, comprising 82% of the disease-causing hvKp isolates[13].

Although hvKp are more prevalent in Asia, infections caused by these organisms have been increasingly reported in other parts of the world[12,14,15]. The US, Canada, and several European countries have documented occasional cases of liver abscess caused by hvKp, mainly associated with travelers or migrants from Asia[16–18]. Few studies have also found hvKp isolates in the gastrointestinal tracts of healthy individuals, with the carriage rate ranging from 4 to 5.2% in China, and 11.3 to 16.7% among healthy people in Malaysia[19,20]. Further molecular investigations have revealed a close genetic relationship between hvKp isolates from human fecal carriage and liver abscess[21,22]. This suggests that the human gastrointestinal tract may act as a reservoir for transmission and infection. The majority of hvKp strains reported to date have shown susceptibility to antibiotics. However, there have been increasing reports on convergent Kp strains across different lineages that display both multidrug resistance (MDR) and virulence. There are two major mechanisms of convergence: hvKp strains acquiring the AMR genes/plasmids as seen in ST23-$bla_{KPC}$[23]ST65-$bla_{CTX-M}$[24] in China, or the MDR cKp strains acquiring the virulence plasmid as observed in ST11-$bla_{KPC}$ in China[25], ST231-$bla_{OXA-232}$ in India[26]. The global emergence of carbapenem-resistant hypervirulent *K. pneumoniae* represents a significant and escalating threat to global health[27].

While there is no consensus definition of hvKp infections, hvKp isolates typically harbor a large IncFIBK plasmid of about 200 kb[28]. This plasmid carries a number of virulence factors that are either absent or present at a low prevalence in cKp isolates, including the siderophores: aerobactin (*iucABCD, iutA*), salmochelin (*iroBCDN*) and the hypermucoid regulator (*rmpA/rmpA2*)[28]. It also possesses several gene clusters encoding toxin-antitoxin systems and resistance to heavy metals, including copper (*pcoABCDRS*), silver (*silABCEPSR*), iron (*fecABDERI*), arsenic (*arsABDR*), and tellurium (*terABCDEFZ*)[14,29,30]. Notably, the virulence plasmid lacks a type IV secretion system and has been experimentally demonstrated to be non-conjugative[30,31]. hvKp often display a hypermucoid phenotype which has been employed to detect hvKp infection via the string test[32]. Nonetheless, the string test fails to identify hvKp isolates lacking the capsule regulator *rmpA/rmpA2* gene, and its accuracy can be affected by culture conditions[33]. Currently, molecular detection of plasmid-encoded virulence genes (*iucA, iroB, rmpA,* and *rmpA2*) is widely considered the most accurate method for distinguishing hvKp from cKp isolates[34,35].

Data from multi-hospital surveillance in Vietnam have shown that Kp are a frequent cause of invasive infections, accounting for 13% of all bacterial isolates from blood and CSF samples[36]. Nonetheless, there are no published data concerning the molecular epidemiology of hvKp infections in this setting. Here, we conducted whole genome sequencing (WGS) and plasmid analysis of hvKp isolates derived from bloodstream infections between 2010 and 2020, as well as two cohorts of healthy individuals, one including children and the other involving adults. Our study provided the first comprehensive insights into the population structure, transmission dynamics, and antimicrobial resistance profiles of hvKp in Vietnam. We further deciphered the genetic structure and horizontal transfer mechanism of the virulence plasmid between Kp isolates.

## Results

### Prevalence and sequence type distribution of hypervirulent *Klebsiella pneumoniae* isolates in Vietnam

From 1 January 2010 to 31 December 2020, 108,303 blood samples were submitted for culture, yielding a positivity rate of 8.8%. Out of the total 700 Kp isolates responsible for bloodstream infections (BSIs) at HTD between 2010 and 2020, 100 isolates (14.3%) were identified as hvKp based on the results of *iucA* PCR. The proportion of hvKp infections varied between 11 and 46 cases per 1000 culture-confirmed BSIs, depending on the year (Fig. S1). We identified 13 hvKp isolates from 350 rectal swabs of healthy adult farmers, representing a prevalence of 3.7 percent. Conversely, no hvKp isolates were found in the 500 rectal swabs of healthy children. Most hvKp isolates (89%, 101/113) displayed a hypermucoid phenotype by the string test.

WGS analysis of 113 hvKp isolates revealed that 111 isolates (98.2%) were identified as *K. pneumoniae*, while the remaining two isolates (1.8%) were *K. quasipneumoniae* subsp. *similipneumoniae*. Remarkably, among 113 hvKp isolates, we identified a diverse array of 17 sequence types (STs). The top three most common STs were ST23 (52%), ST86 (11%), and ST65 (8%), followed by ST420 (4%), ST268 (4%), ST25 (4%), and ST660 (3%). Additionally, there were 12 other STs, each represented by one or two isolates (Table 1). Notably, ST23 (*n* = 57), ST86 (*n* = 9), and ST65 (*n* = 9) were the most common genotypes found in patients with BSIs, while ST268 (*n* = 3), ST23 (*n* = 2), ST86 (*n* = 2), and ST420 (*n* = 2) were prevalent in healthy individuals.

We identified ten distinct K capsular types and seven O antigens. The most common K loci were K1 (51%) and K2 (16%), followed by K20 (10.6%), K5 (8%), K113 (5.3%), and another five K types (9.1%). While K1 was confined to ST23, K2 and K5 were associated with ST86, ST65, and ST25, and K20 was identified in ST420 and ST268. O1 antigen was predominantly found in 87 isolates (77%), spanning various STs: ST23 (*n* = 50), ST86 (*n* = 10), ST65 (*n* = 8), ST420 (*n* = 5), ST25 (*n* = 4). This was followed by the O2 antigen identified in 21 isolates (18.6%) across eight different STs. The remaining O antigens (O3, O4, O5) accounted for 4.4% of isolates (Table 1).

### Phylogenetic relatedness and genetic profile of hypervirulent *Klebsiella pneumoniae* isolates

We reconstructed a phylogenetic tree encompassing 111 hvKp isolates (98 from BSIs and 13 from human carriers) and mapped it against the acquired AMR genes, key virulence factors, and plasmid replicon types (Fig . 1). Overall, each ST formed a discernable phylogenetic cluster, except for ST29 and its single-locus variant ST714, which were inferred as a single cluster. Each phylogenetic cluster encompassed isolates spanning multiple years, suggesting the endemic circulation of various hvKp lineages in Vietnam. Within the phylogenetic tree, hvKp isolates from BSIs and human carriers of the same ST clustered together, demonstrating a shared virulence profile and limited genetic variations (mean pairwise SNP distances ranging from 38 SNPs to 290 SNPs). We further investigated the phylogenetic structure of ST23 isolates in a global context to depict the transmission dynamics of this dominant clone. Our analysis demonstrated that ST23 isolates from Vietnam did not originate from a single common ancestor; instead, they were likely associated with multiple strain introductions. Four phylogenetic clusters (C1–C4) were designated, with the major C4 cluster (virulence gene profile: *iuc1-iro1-clb2-ybt1-rmp1*) spanning from 2011 to 2020, indicating sustained endemic transmission. The C1 and C4 clusters primarily consisted of ST23 isolates from Vietnam, with sporadic isolates from Singapore, Norway, France, and Australia (Fig. S3).

**Table 1 | Genetic profiles of hypervirulent *Klebsiella pneumoniae* isolates from Vietnam**

| Characteristics | Hypervirulent *Klebsiella pneumoniae* (n = 111) | | | | | | |
|---|---|---|---|---|---|---|---|
| | ST23 (n = 59) | ST86 (n = 12) | ST65 (n = 9) | ST420 (n = 6) | ST268 (n = 5) | ST25 (n = 4) | ST Others (n = 16) |
| **Key virulence factors** | | | | | | | |
| Salmochelin: *iro* | 59 | 12 | 9 | 6 | 5 | 4 | 16 |
| Aerobactin: *iuc* | 59 | 12 | 9 | 6 | 5 | 4 | 16 |
| Yersiniabactin: *ybt* | 56 | 8 | 5 | 6 | 5 | 2 | 13 |
| Colibactin: *clb* | 48 | 0 | 5 | 0 | 5 | 0 | 1 |
| Hypermucoid phenotype: *rmpA/A2* | 37 | 7 | 5 | 2 | 3 | 3 | 7 |
| **O antigens** | | | | | | | |
| O1 | 50 | 10 | 8 | 5 | 0 | 4 | 10 |
| O2 | 9 | 1 | 1 | 1 | 5 | 0 | 4 |
| O3 | 0 | 0 | 0 | 0 | 0 | 0 | 2 |
| O4 | 0 | 0 | 0 | 0 | 0 | 0 | 1 |
| **K capsules** | | | | | | | |
| K1 | 56 | 0 | 0 | 0 | 0 | 0 | 0 |
| K2 | 0 | 7 | 7 | 0 | 0 | 3 | 1 |
| K5 | 0 | 4 | 2 | 0 | 0 | 1 | 2 |
| K20 | 0 | 0 | 0 | 6 | 5 | 0 | 1 |
| K113 | 2 | 0 | 0 | 0 | 0 | 0 | 4 |
| Others | 1 | 0 | 0 | 0 | 0 | 0 | 9 |
| **Plasmid replicons** | | | | | | | |
| IncFIBK | 59 | 12 | 9 | 6 | 5 | 4 | 16 |
| IncFII | 5 | 1 | 2 | 0 | 2 | 2 | 2 |
| IncFIB(pKPHS1) | 12 | 0 | 0 | 1 | 0 | 0 | 2 |
| IncR | 2 | 0 | 0 | 0 | 0 | 1 | 0 |
| IncA/C | 0 | 0 | 1 | 0 | 0 | 0 | 0 |
| IncN | 0 | 1 | 0 | 0 | 0 | 0 | 0 |

Genomic analysis revealed that all isolates possessed at least two plasmid-encoded siderophores biosynthesis genes: *iuc* (aerobactin) and *iro* (salmochelin). Additionally, 99 isolates (89%) harbored the hypermucoid regulator *rmpA/rmpA2* gene and 95 isolates (86%) contained the chromosomal yersiniabactin locus (*ybt*). More than half of the isolates (53%) also carried the *clb* locus encoding colibactin (Table 1). The distribution of key virulence factors displayed distinct patterns both within and between STs. The majority of ST23 isolates (48/59, 81%) displayed the *iro1-iuc1-rmpA1*(+/-)*-ybt1-clb2* virulence pattern, while the remaining isolates (11/59, 19%) exhibited the *iro1-iuc1-rmpA1-ybt8* pattern (Fig. 1). Alternatively, seven out of 11 ST86 isolates contained the yersiniabactin-encoding *ybt10/22/9* locus but lacked the colibactin-encoding locus; whereas, five out of nine ST65 isolates carried both the yersiniabactin and colibactin-encoding loci (*ybt17-clb3*). Most hvKp strains (108/111) harbored the *iro1-iuc1* virulence profile, with the exception of three isolates carrying the plasmid-encoded *iuc3* and the chromosomal ICEKp1 (2 ST25) and IS100 (1 ST828) *iro3*.

Among 111 hvKp isolates, 108 contained a closely related InFIBK +/−IncHI1B virulence plasmid, exhibiting significant genetic similarity (73.8–99.9% coverage) to the canonical virulence plasmid pLVPK in hvKp. Conversely, three isolates harbored an IncFIBK-like plasmid with low coverage (ranging from 38.1 to 43.3%). The plasmid profiling and long-read Nanopore sequencing data confirmed the presence of additional plasmids in hvKp isolates, including IncFII+/-IncFIA (10%), IncR (2.7%), IncA/C (0.9%) plasmid and IncFIB phagemid (13.5%). While the majority of hvKp isolates displayed pan-susceptibility to the tested antibiotics and lacked AMR genes, 14 isolates (12.6%) were identified to carry at least one AMR gene. These isolates were obtained from both BSIs and human carriers, encompassing different STs (Table 2). Two of them harbored an AMR gene on the chromosome, including one ST23 isolate (ISEc9-*bla*CTX-M-14) and one ST86 isolate (ISKpn74-*sul1*). Five

isolates carried a similar array of AMR genes (*qnrS1-sul-tetA-dfrA1*(+/−)*-bla*LAP-2(+/−)) on an IncFII+/−IncFIA plasmid. These IncFII plasmids showed genetic variability and were distributed among different STs: ST268 (2), ST23 (1), ST828 (1), and ST25 (1). One ST65 isolate acquired an IncA/C plasmid of 246 kb, carrying 14 different AMR genes (*aadA24-aadB-arr2-bla*CTX-M*-cmlA5-dfrA23-bla*LAP-2*-bla*OXA-10*-qnrS1-floR-strA-sul2-tetA-bla*VEB-1). Two isolates carried an IncR plasmid with different AMR genes: *qnrS1-bla*LAP-2*-aac3-IId* (ST23 isolate) and *dfrA1-qnrB4-sul1* (ST25 isolate). The AMR genes were also located in a small col plasmid of an ST25 isolate, IncFIBK-like plasmids of two other ST25 isolates, and two plasmids with unknown replicon types in two ST23 isolates.

## Genetic structure of *iuc3*-carrying virulence plasmids

We sought to determine the genetic structure of the three IncFIBK-like plasmids (designated as pQY1, pQY2, and pQY3) identified in two ST25 isolates and one ST828 isolate. Plasmid nanopore sequencing data revealed that these are multi-replicon IncFIBK-IncFII plasmids, carrying gene features from both the IncFIBK virulence plasmid and IncFII plasmid. The IncFIBK virulence region encompassed the aerobactin biosynthesis (*iucABCD-iutA*) and heavy metal resistance operons (copper: *pcoABCDRS*, silver: *silABCEFPRS*, iron: *fecABCDERI*, and arsenic: *arsABDR*). Alternatively, the IncFII region of the *iuc3*-carrying plasmids contained *tra* operon encoding for type IV conjugation machinery, with or without AMR gene cassettes (Fig. 2).

The three *iuc3*-carrying plasmids shared a conserved region of about 139.8 kb (pairwise nucleotide identity >99%) while exhibiting differences in size and gene content. The pQY1 plasmid was 240.6 kb in length, carrying a set of AMR genes, *tetA-sul2-strAB-dfrA14-AadA1-bla*OXA-10*-floR-cmlA-arr*. The pQY2 was 194 kb in length and did not carry any AMR genes. The pQY3 was 198 kb and carried *tetA-sul2-strAB-floR-aph6-bla*LAP-2 (Table 3). Furthermore, plasmid comparative

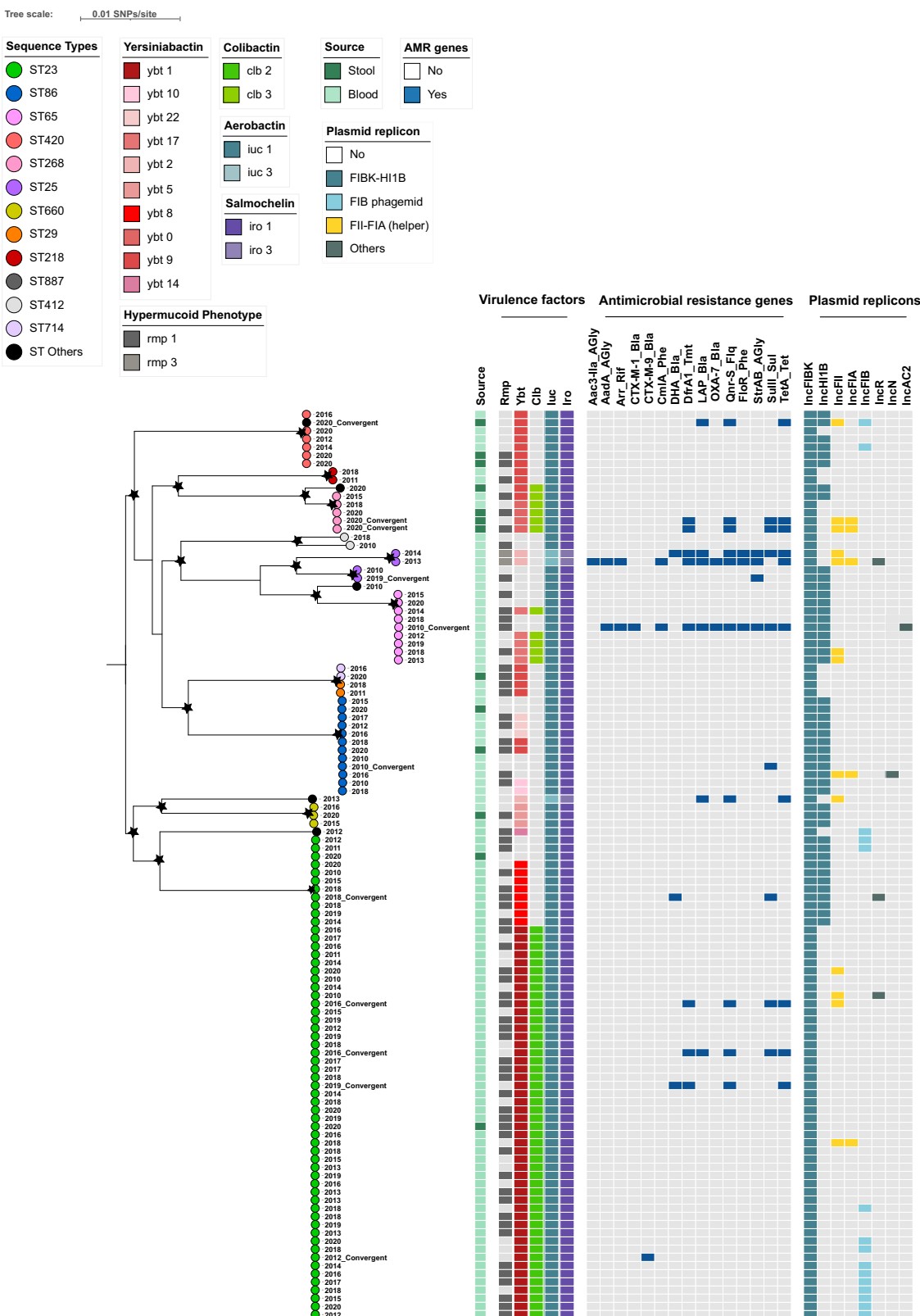

**Fig. 1 | Phylogenetic structure of 111 hypervirulent *Klebsiella pneumoniae* isolates from Vietnam between 2010 and 2020.** The phylogenetic tree is mid-point rooted. The sequence types (STs) are represented by the colored circles at the terminal nodes of the tree. The years of isolation are shown on the tip labels. The tree scale bar represents the number of SNPs per site. The black stars highlight bootstrap support values ≥80% on internal nodes, with larger stars indicating higher bootstrap values. The heat maps display the source of isolation (bloodstream infections, human carriage), followed by the presence (in color) or absence (in gray) of different alleles of virulence genes, resistance genes, and plasmid replicon types. The hvKp isolates exhibiting convergent phenotype (AMR and virulence) or carrying the *iuc3* plasmids are indicated at the tip labels. Source data for the heat maps are provided as a Source Data file.

**Table 2 | Genetic characteristics of 14 convergent *K. pneumoniae* isolates**

| Source | ST | AMR gene profile | AMR gene location | AMR plasmid | Virulence plasmid |
|---|---|---|---|---|---|
| Blood | ST86 | Sul1 | Chromosome | - | IncFIBK:IncHI1B |
| Blood | ST23 | bla_CTX-M-14 | Chromosome | - | IncFIBK |
| Blood | ST25 | StrA-Sul2 | Plasmid | Col | IncFIBK:IncHI1B |
| Blood | ST65 | StrA-AadA24-AadB-Arr2-CmlA5-FloR-bla_CTX-M-15-bla_LAP-2-bla_VEB-1-bla_OXA-10-QnrS1-Sul2-TetA-DfrA23 | Plasmid | IncAC | IncFIBK:IncHI1B |
| Stool | ST268 | QnrS1-Sul1-TetA-DfrA1 | Plasmid | IncFII:FIA | IncFIBK:IncHI1B |
| Blood | ST23 | QnrS1-Sul1-TetA-DfrA1 | Plasmid | IncFII | IncFIBK |
| Blood | ST828 | bla_LAP-2-QnrS1-Sul2-TetA | Plasmid | IncFII | IncFIBK |
| Blood | ST25 | StrA-FloR-bla_DHA-1-bla_LAP-2-QnrS1-Sul2-TetA-DfrA1 | Plasmid | IncFII plus IncFIBK-IncFII | IncFIBK |
| Stool | ST268 | QnrS1-Sul1-TetA-DfrA1 | Plasmid | IncFII | IncFIBK:IncHI1B |
| Stool | ST1544 | aac(3)-IV-bla_LAP-2-QnrS1-Sul2-TetA | Plasmid | IncFII | IncFIBK:IncHI1B |
| Blood | ST23 | bla_DHA-1-QnrB4-Sul1 | Plasmid | IncR | IncFIBK:IncHI1B |
| Blood | ST23 | bla_DHA-1-QnrS1-TetA-DfrA1 | Plasmid | Unknow replicon | IncFIBK |
| Blood | ST23 | bla_LAP-2-QnrS1-Sul1-TetA-DfrA1 | Plasmid | Unknow replicon | IncFIBK |
| Blood | ST25 | StrA-Aac3-Iid-AadA1-Arr2-CmlA5-FloR-bla_LAP-2-bla_OXA-10-QnrS1-Sul2-TetA-DfrA1 | Plasmid | IncFIA-IncR plus IncFIBK:IncFII | IncFIBK:IncFII |

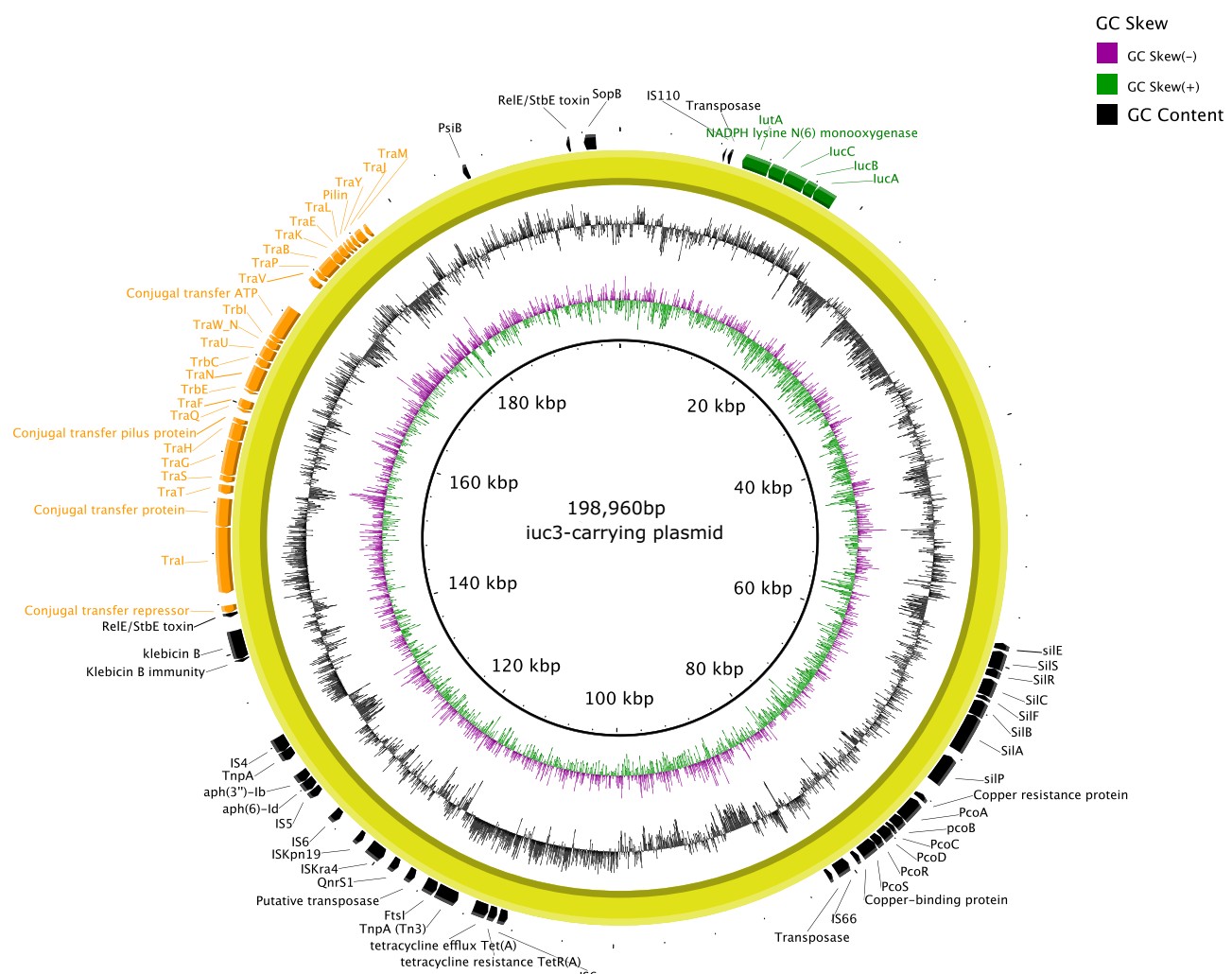

**Fig. 2 | Genetic structure of *iuc3* virulence plasmid in a hypervirulent *K. pneumoniae* isolates from Vietnam.** The *iuc3* virulence plasmid pQY3 is represented by a yellow circle. The GC content and GC skew of the pQY3 plasmid are displayed in the two innermost rings. The gene annotations are displayed as arrows in the outermost ring. The conjugation module is indicated in orange, while the virulence genes are highlighted in green.

**Table 3 | Key genetic features of full-length plasmids in hypervirulent *Klebsiella pneumoniae* isolates**

| ID | ST | Plasmid | Inc type | Size (bp) | Virulence genes | Tra genes | T4CP | Relaxase | oriT | AMR genes | Tellurite resistance genes |
|---|---|---|---|---|---|---|---|---|---|---|---|
| 65 | ST23 | Helper | IncFII | 74.503 | - | TraX-D-G-H-B-Q-F-N-U-W-C-V-K-E-L-A-Y-M | + | + | + | - | - |
| | | Phagemid | FIB | 106.291 | - | - | + | - | - | - | - |
| 75 | ST86 | Virulence | IncFIBK:HIB | 195.619 | iucABCD-iutA-iroBCDN-rmpA/A2 | - | + | - | - | - | + |
| | | Helper | IncFII:FIA:N | 121.952 | - | TrwJ-K-M-L-N-D-E-F-G-I,TraD-A-M-L-E-K-B-V-C-W-U-N-F-Q-H-G-D-X | + | + | + | - | - |
| 78 | ST23 | Virulence | IncFIBK:HIB | 230.437 | iucABCD-iutA-iroBCDN-rmpA/A2 | - | + | - | + | - | + |
| | | Helper | IncFII | 95.989 | - | TraM-A-L-E-K-B-V-C-W-U-N-F-Q-H-G-D-X | + | + | + | DfrA1, QnrS, SulI, TetA | - |
| 88 | ST65 | Virulence | IncFIBK:HIB | 240.534 | iucABCD-iutA-iroBCDN-rmpA/A2 | - | + | - | + | - | + |
| | | Helper | IncA/C:N | 246.199 | - | TraM-Y-A-L-E-K-B-V-C-W-U-N-F-Q-H-G-D-X | + | + | + | AadAB, Arr2, CTX-M-15, CmlA, DfrA, LAP, OXA-10, QnrS, FloR, StrAB, SulII, TetA, VEB-1 | - |
| 121 | ST23 | Virulence | IncFIBK:HIB | 223.585 | iucABCD-iutA-iroBCDN-rmpA/A2 | - | + | - | + | - | + |
| | | Phagemid | FIB | 109.376 | - | - | + | - | - | - | - |
| | | Helper | Unknow | 176.873 | - | TraN-G-H-F-U-W-C-V-B-K | + | + | - | - | - |
| 146 | ST25 | Virulence (iuc3) | IncFIBK:FII | 241.139 | iucABCD-iutA | TraM-A-L-E-L-B-V-C-W-U-N-F-Q-H-G-D-X | + | + | + | TetA,Sul2,StrAB,DfrA14,AadA1,OXA-10,FloR,CmlA,Arr | - |
| | | AMR | IncFIA:R | 82.512 | - | - | - | - | + | QnrS, LAP, Aac3 | - |
| 201 | ST828 | Virulence (iuc3) | IncFIBK:FII | 194.876 | iucABCD-iutA | TraX-D-G-H-B-Q-F-N-U-W-C-V-K-E-L-A-M | + | + | + | - | - |
| | | AMR | IncFII | 82.668 | - | TrbC-A-B, TraY-X-W-U-T-R-Q-P-O-N-M-L-K-J-I-H | + | + | + | LAP, QnrS, SulII, TetA | - |
| 206 | ST25 | Virulence (iuc3) | IncFIBK:FII | 198.96 | iucABCD-iutA | TraX-I-D-G-H-B-Q-F-N-U-W-C-V-K-E-L-A-Y-M | + | + | + | TetA,QnrS,LAP-2,APH-(6),Sul2,StrAB,FloR | - |
| | | AMR | IncFII | 77.321 | - | TraO-P-Q-R-T-U-W-X-Y, TrbC-B-A | + | + | + | SulII, DfrA, DHA, QnrS, TetA | - |
| 176 | ST1049 | Virulence | IncFIBK | 299.039 | iucABCD-iutA-iroBCDN-rmpA/A2 | - | + | - | + | - | + |
| | | Phagemid | FIB | 111.709 | - | - | + | - | - | - | - |
| 410 | ST420 | Virulence | IncFIBK:HIB | 219.173 | iucABCD-iutA-iroBCDN-rmpA/A2 | - | + | - | + | - | + |
| | | Phagemid | FIB | 112.412 | - | - | + | - | - | - | - |

analysis with five publicly available *iuc3*-carrying plasmids derived from hvKp isolates in Thailand, Laos PDR, and the USA revealed a high degree of genetic similarity (identity: 99.7–99.9%, coverage: 60–97%) (Fig. S4). The *iuc3* plasmid from Thailand (CP041094) originated from a pig, while the two *iuc3* plasmids from the US (CHS48 and CHS43) were found in two hvKp ST25 isolates. Notably, one *iuc3* plasmid from a previously published hvKp strain in Vietnam (p130411) was almost identical to the pQY1 plasmid (identity: 99.9%, coverage: 100%).

### Horizontal transfer mechanism of the non-conjugative virulence plasmid

Although the IncFIBK virulence plasmid is not self-transferable due to the lack of T4SS-encoding *tra* gene cluster, it was found across various sequence types of *K. pneumoniae* as well as *K. quasipneumoniae* isolates in this study. We aimed to gain insights into the transmission of the virulence plasmid by examining the congruence between the chromosomal and plasmid phylogenies and conducting a series of conjugation experiments. Apart from the three *iuc3*-encoding IncFIBK-IncFII virulence plasmids, all 108 *iuc1*-encoding IncFIBK+/−IncHIB virulence plasmids displayed a high synteny with the reference pLVPK plasmid, with the mapping coverage ranging from 73.8 to 99.9% (Fig. 3). The resulting plasmid phylogeny showed distinct clusters with strong bootstrap support values. We observed several incongruences between the chromosomal and plasmid phylogenies; for instance, ST660 and ST65 formed separate clusters on the chromosomal phylogeny but grouped together on the plasmid phylogeny. Similar discrepancies were identified between ST25 and ST420, as well as between ST218, ST29, ST714, and ST412 (Figs. 1, 3). These findings provided a strong indication that the virulence plasmid has been horizontally transferred across genotypes and *Klebsiella* species. In contrast, for ST23 and ST86, congruence in the chromosomal and plasmid phylogenetic structures of isolates suggests the virulence plasmid has co-evolved with these host bacteria for some time.

Next, we investigated the horizontal transfer ability of the virulence plasmid between hvKp and cKp via conjugation experiments. Thirty-three hvKp donor strains were chosen to represent all different sequence types and the diversity of non-virulence plasmids. Our conjugation experiments demonstrated the 3 *iuc3*-encoding IncFIBK:IncF plasmids with intact T4SS were successfully transferred to the recipient Kp strain. Out of 30 hvKp isolates carrying the *iuc1*-encoding IncFIBK virulence plasmid, five isolates were capable of transferring the virulence plasmid to the recipient Kp strain (Fig. 4). Plasmid Nanopore sequencing confirmed that these isolates contained a conjugative helper plasmid, including an IncFII plasmid with sizes ranging 74 kb to 176 kb (3 ST23 isolates), an IncFII:IncFIA:IncN plasmid of 122 kb (one ST86 isolate) and an IncA/C:IncN plasmid of 246 kb (one ST65 isolate). Furthermore, the predicted *nic* site (10 bp) in the oriT region of these conjugative helper plasmids exhibited significant sequence similarity with the *nic* site of the IncFIBK virulence plasmid (Table 4 and Fig. 5). The *nic* site is a cleaving site of relaxase, a DNA transferase that catalyzes a strand- and sequence-specific cleavage and initiates the plasmid conjugal transfer. This finding signifies that when hvKp isolates acquire a conjugative helper plasmid with a shared *nic* site in the oriT, the non-conjugative virulence plasmid within these hvKp isolates can be effectively transferred to other classical Kp strains. This notion was further substantiated by the observation that the virulence plasmid remained non-transferable when the hvKp isolates (*n* = 22) lacked a conjugative helper plasmid or when the hvKp isolates (*n* = 3) harbored a conjugative helper plasmid with dissimilar *nic* site regions (Table 4 and Fig. 5). Moreover, by comparing the plasmid profiles of donor, recipient and the randomly selected transconjugants, we found that the virulence plasmid was either transferred alone, or co-transferred with another plasmid (conjugative helper plasmid/phage-like plasmid), or formed a hybrid plasmid with the conjugative helper plasmid (Fig. 4).

## Discussion

Hypervirulent *Klebsiella pneumoniae* have emerged as a significant cause of severe community-acquired infections in Asia, with evidence of its global spread[20,37,38]. Despite its significance, the epidemiological and genetic landscape of hvKp infections in LMICs have been poorly characterized due to the lack of reliable detection methods and comprehensive molecular investigations. This study offers a multiplex real-time PCR assay with melting curve analysis for detecting hvKp isolates based on their typical gene markers, given the absence of a consensus definition for clinical infection. Furthermore, the hvKp isolates identified by PCR showed 100% sensitivity when compared to WGS. This assay can be readily employed in LMICs to identify hvKp infections for rigorous clinical examinations and effective treatment, as well as epidemiological investigations. Given the scarcity of data regarding hvKp infections in Vietnam, our work was first to describe the prevalence, population structure, and virulence plasmid transfer of hvKp in this setting. Akin to previous studies, the preponderance of hvKp infections in Vietnam was attributed to the three prevailing lineages of ST23-K1, ST86-K2, and ST65-K2. Nevertheless, the population structure of hvKp isolates displayed significant diversity, with a multitude of distinct lineages co-circulating throughout the study period. Moreover, hypervirulent *Klebsiella* infections were not exclusive to *K. pneumoniae* but were also attributed to two *K. quasipneumoniae* subsp. *similipneumoniae* isolates (ST367 and ST816), all of which shared a common virulence plasmid. These findings raise significant concerns regarding the emergence and international spread of novel non-K1 and non-K2 hypervirulent *Klebsiella* strains, which may exhibit distinct propensities for acquiring antimicrobial resistance genes or differing capabilities in pathogenesis and transmission[39,40]. The heterogeneous makeup of surface structures, including K capsular types and O antigens, within the hvKp strains also presents a substantial challenge in the development of preventive measures, such as vaccines and innovative therapeutics, targeting these surface antigens. Alternatively, protein-based approaches should be considered for vaccine development as they have the potential to offer broader coverage across various *Klebsiella* strains.

Previous studies have shown little evidence of *K. pneumoniae* transmission between humans and non-human sources[41–43]. Meanwhile, hvKp have been found in the gastrointestinal tracts of both healthy individuals and patients[19,44]. Here, we sought to understand the circulation of hvKp strains in healthy individuals and the extent of genetic sharing with disease-causing organisms. In healthy children, while classical *K. pneumoniae* strains were identified from rectal swabs, no hvKp strains were detected. We speculate this observation may stem from the frequent exposure of children in our setting to antibiotics[45,46], coupled with the prevailing pan-susceptibility of most hvKp strains to these antimicrobial agents. In healthy adults, we found a prevalence of 3.7% of hvKp isolates, exhibiting diverse but significant overlap in sequence types with those found in humans with bloodstream infections. The fact that the carriage and infecting isolates share the same STs and exhibit limited genetic distances strongly suggests that the human gastrointestinal tract constitutes an important reservoir for the transmission of hvKp strains. The biological and epidemiological factors governing the establishment of gut colonization in susceptible hosts and the community transmission of hvKp remain poorly understood. We need to address these knowledge gaps to devise appropriate preventative measures to prevent/reduce risks of gut colonization and infection development in vulnerable populations. Furthermore, although hvKp infections have been rarely reported in children, it is essential to note that such infections can lead to severe illnesses with high mortality rates[47]. Given that our study primarily centered on bloodstream infections in adult patients, we strongly advocate for further investigations into the community transmission and incidence rates of various hvKp infections across diverse populations (adults vs. children) and settings (community vs. hospital environments).

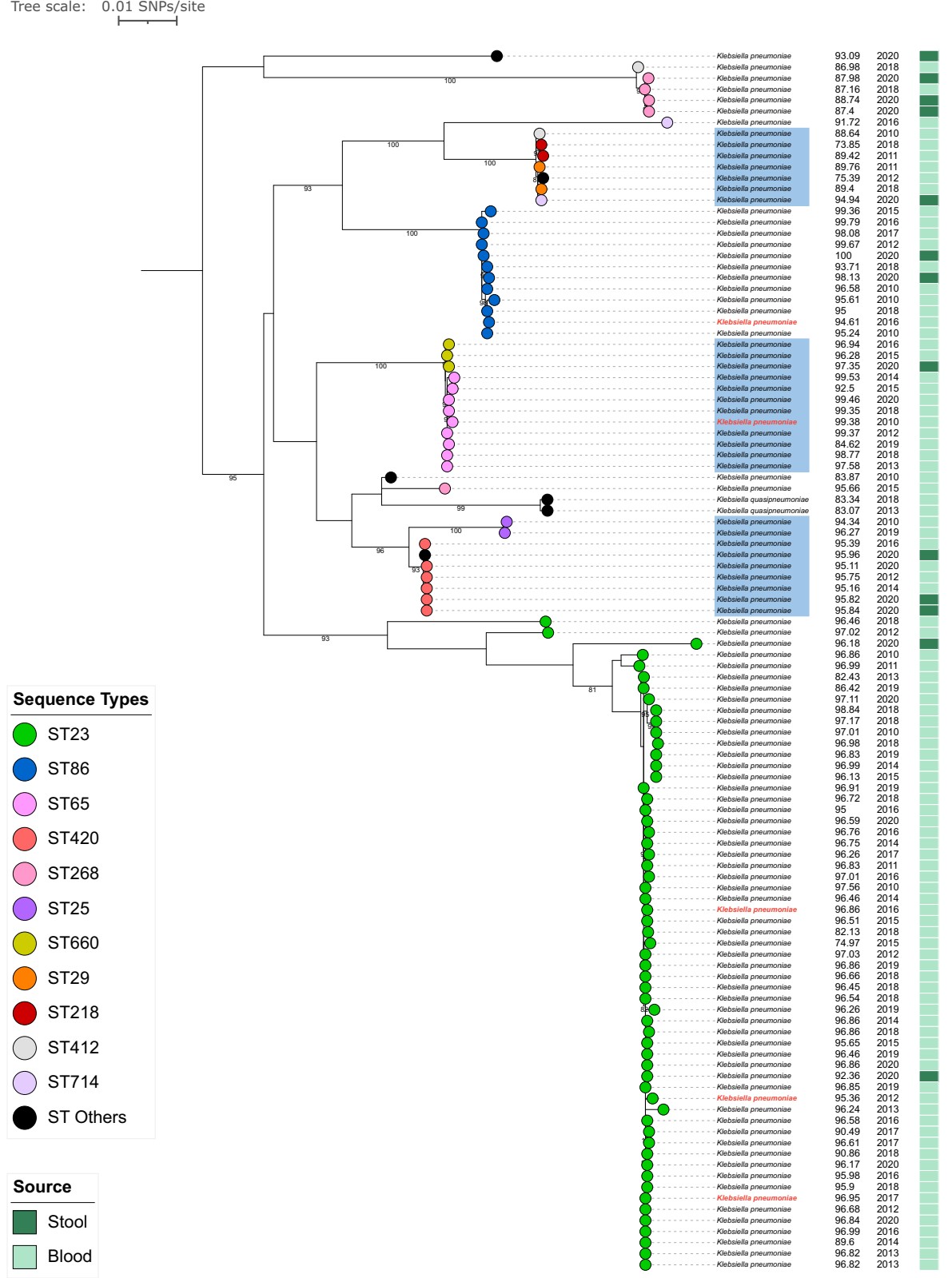

**Fig. 3 | The plasmid phylogeny of *iuc1* virulence plasmids in hypervirulent *Klebsiella* isolates from Vietnam.** The plasmid phylogenetic tree is mid-point rooted. The sequence types (STs) of isolates are represented by the colored circles at the terminal nodes of the tree. The virulence plasmids that are successfully transferred into the *Klebsiella pneumoniae* recipient strain via conjugation are marked in red at the tip labels. The blue shaded boxes highlight STs that are associated with incongruences between the chromosomal and plasmid phylogenies. The tree scale bar represents the number of SNPs per site. The numbers on the branches indicate nodes with bootstrap support values ≥ 80%. The first column shows the percentage of mapping coverage against the reference virulence plasmid. The second column indicates the year of isolation, and the third columns show the source of isolation (light green: blood, dark green: stool). Source data are provided as a Source Data file.

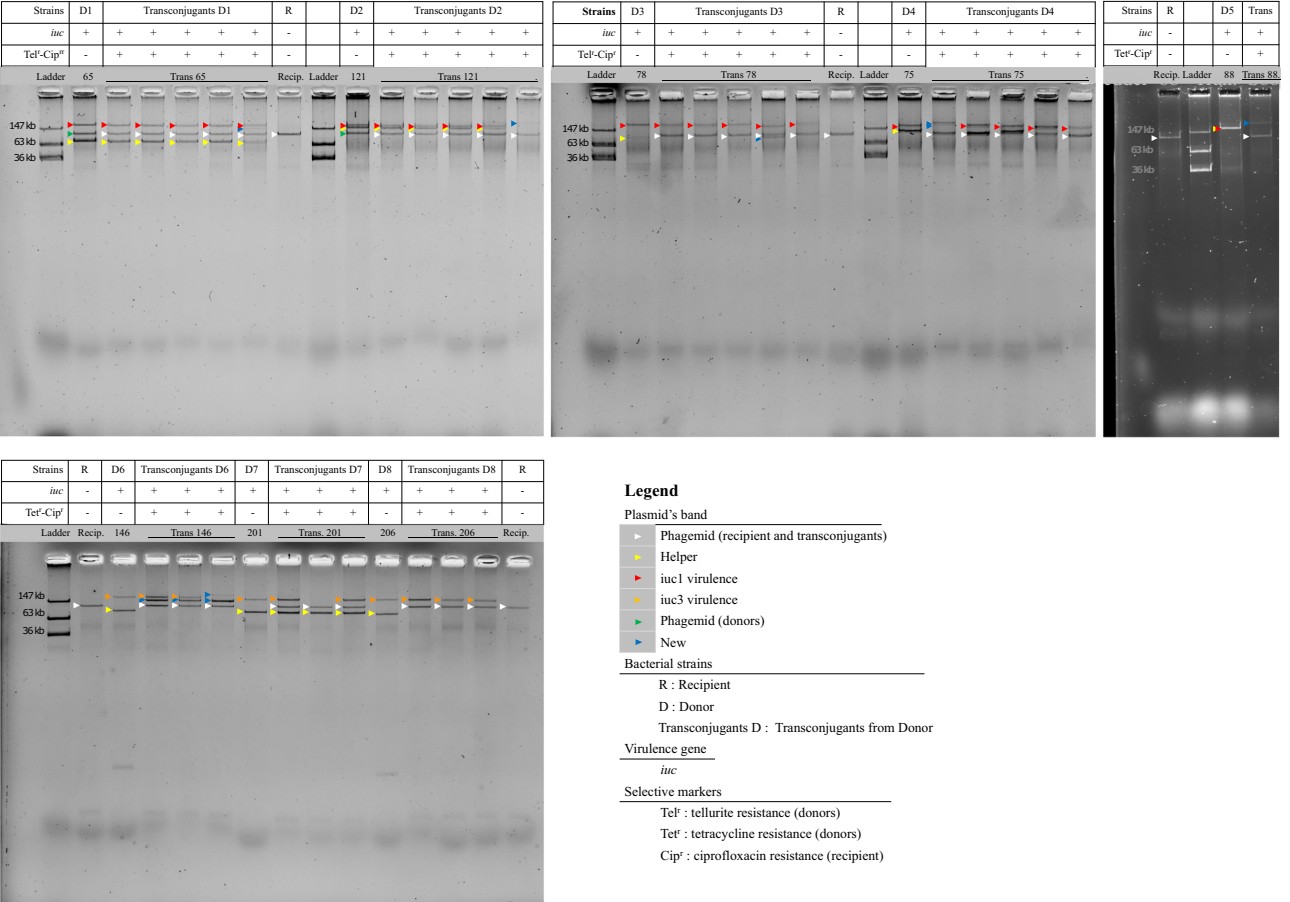

**Fig. 4 | Plasmid profiling of transconjugants containing the virulence plasmid.** This figure illustrates the plasmid profiles of the selective donors (D1–D8), recipient (R), and transconjugants (Transconjugants D1–D8), as well as the confirmation results of successful plasmid transfer by qPCR (*iuc* gene) and selective media. The arrows are colored according to different plasmids, including white: phagemid (recipient and transconjugants); yellow: helper plasmid, red: *iuc1* virulence plasmid, orange: *iuc3* virulence plasmid, green: phagemid (donors); and blue: virulence

plasmids in transconjugants that differ in sizes compared to the original virulence plasmids in donors. Selective markers: Tellurite resistance (Tel^r), Tetracycline resistance (Tet^r), and Ciprofloxacin resistance (Cip^r) are represented with (+/−) to denote the growth or no growth of bacterial strains on selective media. The conjugation experiments were repeated twice with similar results. Source data are provided as a Source Data file.

Although most hvKp isolates were susceptible to antibiotics, the convergence of AMR and virulence genes was observed in 12.6% of isolates. The convergence events were also observed across different STs. The occurrence of MDR-hvKp strains is very concerning as these organisms may gain a competitive advantage and spread in settings with strong antibiotic selection pressures. This trend needs to be closely monitored as such emergence appears unavoidable and could result in substantial global health threats[38,48,49]. A significant finding from our research lies in the identification of specific conjugative helper plasmids, not only contributing to the MDR phenotype but also offering a means for the horizontal transfer of the non-conjugative virulence plasmid between *K. pneumoniae* isolates. These helper plasmids shared a homologous oriT region with the virulence plasmid and predominantly belonged to the IncFII group, which is common in human commensal Enterobacteriaceae[50,51]. Similar findings were reported recently, showing that a *bla*_KPC-positive IncFII plasmid sharing the *nic* site in the oriT with the virulence plasmid can help to mobilize the virulence plasmid from hvKp to cKp strains[49]. We surmise that hvKp continues to evolve antibiotic resistance while spreading the virulence phenotype to other niche-sharing classical *Kp* strains through the acquisition of conjugative helper plasmids. Strikingly, we observed that during the conjugation via the helper plasmids, plasmid profiling showed the differing sizes of virulence plasmids in the transconjugants. This is probably the result of recombination or

co-integration between the virulence plasmid and the helper plasmid. Such recombinant/co-integrative events were observed previously, resulting in the formation of hybrid plasmids[52,53]. These hybrid plasmids can carry a combination of virulence factors, AMR genes, and T4SS, which may facilitate its further spread while reducing the fitness cost associated with the virulence plasmid[6]. The high prevalence of MDR bacteria in settings like Vietnam may act as a significant gene pool, contributing to the emergence of novel hvKp and MDR-hvKp strains.

In this study, we also found three *iuc3*-carrying IncFII:IncFIBK virulence plasmids belonging to ST25 and ST828. These plasmids were self-transferable and shared little coverage with the *iuc1*-carrying IncFIBK virulence plasmid identified in most hvKp isolates. Apart from the aerobactin biosynthesis gene cluster, the *iuc3*-carrying virulence plasmids commonly contained several AMR gene cassettes and an intact T4SS but lacked *rmpA* and *iroABCD* genes. The majority of previously characterized hvKp isolates carrying the *iuc3* IncFII:IncFIBK virulence plasmids was traced back to animal sources, thus hinting at a zoonotic reservoir for this plasmid[54,55]. Notably, ST25 over-represented the hvKp isolates that had been shown to harbor the *iuc3*-carrying virulence plasmids[56,57], suggesting that ST25 is a major vehicle for the transmission of the *iuc3* IncFII:IncFIBK virulence plasmid. Here, we found an identical *iuc3* IncFII:IncFIBK virulence plasmid in two different STs (ST25 and ST828) of hvKp strains in Vietnam. This finding

**Table 4 | Origin of transfer (oriT) regions in virulence plasmids and helper plasmids among selective hypervirulent *Klebsiella pneumoniae* isolates for conjugation**

| ID | Sequence Type | Plasmid | Inc type | oriT type | nic site | Transferability of virulence plasmid |
|----|---------------|---------|----------|-----------|----------|--------------------------------------|
| ERS2489075 | ST23 | Virulence | IncFIBK:HIB | oriT_100998 | AGTTTGGTGC | Yes |
|  |  | Helper | IncFII | oriT_100096 | GGTGTGGTGA |  |
| 32_merged | ST86 | Virulence | IncFIBK:HIB | oriT_Q2 (novel) | GGATGTACGA | Yes |
|  |  | Helper | IncFII:FIA:N | oriT_100015 | GCATTTACGA |  |
| 33 | ST23 | Virulence | IncFIBK:HIB | oriT_100998 | AGTTTGGTGC | Yes |
|  |  | Helper | IncFII | oriT_100096 | GGTGTGGTGA |  |
| ERS2489080 | ST65 | Virulence | IncFIBK:HIB | oriT_100998 | AGTTTGGTGC | Yes |
|  |  | Helper | IncA/C:N | oriT_100096 | GGTGTGGTGA |  |
| 47_merged | ST23 | Virulence | IncFIBK:HIB | oriT_100998 | AGTTTGGTGC | Yes |
|  |  | Helper | IncFII | oriT_Q1 (novel) | GAGTTTGCTTG |  |
| 2 | ST23 | Virulence | IncFIBK:HIB | oriT_100998 | AGTTTGGTGC | No |
|  |  | Helper | IncFII:R | oriT_100015 | GCATTTACGA |  |
| 85 | ST23 | Virulence | IncFIBK:HIB | oriT_100998 | AGTTTGGTGC | No |
|  |  | Helper | IncFII | oriT_100015 | GCATTTACGA |  |
| 50 | ST23 | Virulence | IncFIBK:HIB | oriT_100998 | AGTTTGGTGC | No |
|  |  | Helper | IncR | - | - |  |
| 69 | ST23 | Virulence | IncFIBK:HIB | oriT_100998 | AGTTTGGTGC | No |
| 6 | ST25 | Virulence | IncFIBK:HIB | oriT_100998 | AGTTTGGTGC |  |
| 60 | ST29 | Virulence | IncFIBK | oriT_100998 | AGTTTGGTGC | No |
| ERS2489033 | ST367 | Virulence | IncFIBK:HIB | oriT_100998 | AGTTTGGTGC |  |
| 71 | ST412 | Virulence | IncFIBK | oriT_100998 | AGTTTGGTGC | No |
| 18 | ST420 | Virulence | IncFIBK:HIB | oriT_100998 | AGTTTGGTGC |  |
| 11 | ST420 | Virulence | IncFIBK:HIB | oriT_100998 | AGTTTGGTGC | No |
| 27 | ST65 | Virulence | IncFIBK:HIB | oriT_100998 | AGTTTGGTGC |  |
| 8 | ST65 | Virulence | IncFIBK:HIB | oriT_100998 | AGTTTGGTGC | No |
| 34 | ST714 | Virulence | IncFIBK | oriT_100998 | AGTTTGGTGC |  |
| 44 | ST816 | Virulence | IncFIBK:HIB | - | - | No |
| 52 | ST86 | Virulence | IncFIBK:HIB | oriT_100998 | AGTTTGGTGC |  |
| 9_ | ST1049 | Virulence | IncFIBK | oriT_100998 | AGTTTGGTGC | No |
| 91 | ST420 | Virulence | IncFIBK:HIB | oriT_100998 | AGTTTGGTGC |  |
| 66 | ST218 | Virulence | IncFIBK | oriT_100998 | AGTTTGGTGC | No |
| 63 | ST23 | Virulence | IncFIBK:HIB | oriT_100998 | AGTTTGGTGC |  |
| 7 | ST23 | Virulence | IncFIBK:HIB | - | - | No |
| 62 | ST23 | Virulence | IncFIBK:HIB | oriT_100998 | AGTTTGGTGC |  |
| 55 | ST23 | Virulence | IncFIBK:HIB | oriT_100998 | AGTTTGGTGC | No |
| ERS2489056 | ST23 | Virulence | IncFIBK:HIB | oriT_100998 | AGTTTGGTGC |  |
| ERS2489022 | ST23 | Virulence | IncFIBK:HIB | oriT_100998 | AGTTTGGTGC | No |
| ERS2489005 | ST23 | Virulence | IncFIBK:HIB | oriT_100998 | AGTTTGGTGC |  |

suggests that once acquired, the *iuc3* plasmids can spread further among Kp strains. Furthermore, we found that the hvKp strains carrying the *iuc3* virulence plasmids were not hypermucoid due to the absence of the *rmpA* gene or carried the truncated *rmpA*, and thus the string test might fail to detect these organisms as hvKp. The lack of hypermucoid phenotype may also alleviate constraints on horizontal gene transfer, promoting DNA uptake in these hvKp strains[58].

In conclusion, we present the first genomic description of hypervirulent *K. pneumoniae* from bloodstream infections and healthy carriage in Vietnam. Our findings reveal a genetically diverse population of seventeen hvKp genotypes, dominated by ST23, ST86, and ST65. We uncovered the emergence of antimicrobial resistance across different hvKp genotypes largely via the acquisition of AMR plasmids/genes. Strikingly, we found that the acquisition of AMR conjugative helper plasmids sharing the common oriT region with the virulence plasmid facilitated the horizontal transfer of the virulence plasmid between *K. pneumoniae* strains. Carriage and infecting organisms

clustered phylogenetically, indicating human intestinal carriage is likely to play a major role in the transmission of hvKp in the community. Enhanced community surveillance to understand factors driving the establishment of intestinal carriage and facilitating the transmission of hypervirulent *K. pneumoniae* is instrumental to identify appropriate preventative and control measures. Furthermore, molecular testing should be applied in clinical settings to rapidly identify infections caused by hypervirulent *K. pneumoniae* for thorough evaluation and effective management.

## Methods
### Bacterial and data collection
A total of 700 Kp isolates were yielded from a retrospective study of bloodstream infections at the Hospital for Tropical Diseases (HTD) between 2010 and 2020. These isolates were identified from routine blood cultures and stored at the Microbiology Department. For blood culture, an aliquot of 5–8 mL (for adults) or 2–5 mL (for children) of

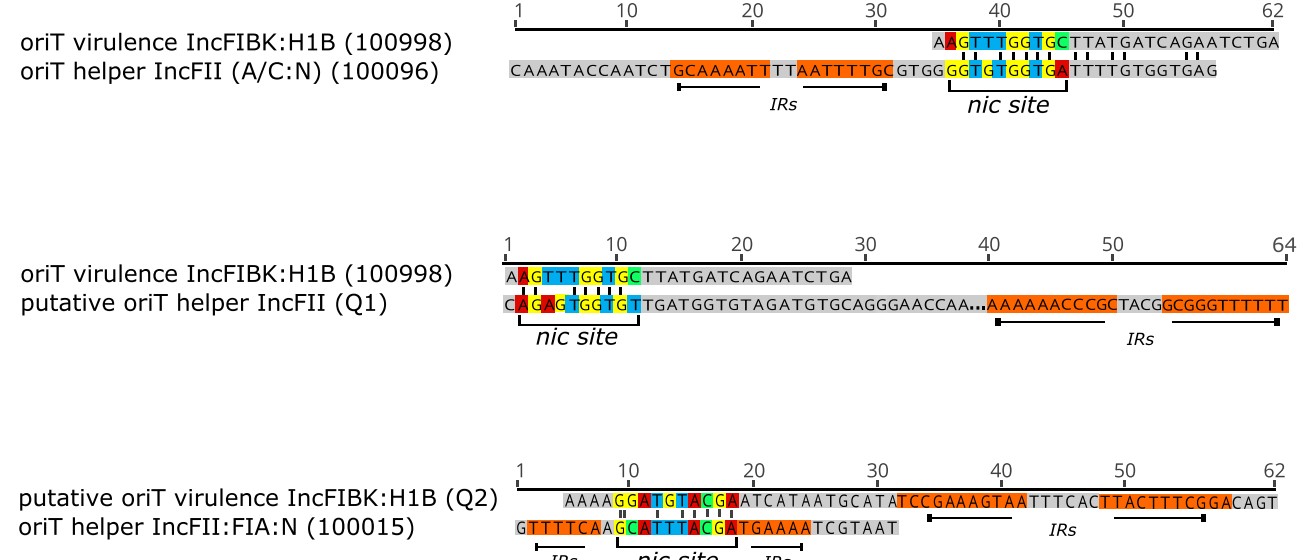

**Fig. 5 | Comparison of the *nic* site sequences between the virulence plasmids (IncFIBK:IncH1B) and selective helper plasmids (IncFII, IncFII:IncFIA:IncN, and IncA/C:IncN).** The inverted-repeat sequences close to the *nic* site of the plasmids are shown by horizontal arrows and block text highlighted in orange. The *nic* site are indicated by nucleotides highlighted in different colors, with red representing nucleotide for "A," blue for "T," yellow for "G," and green for "C".

venous blood were inoculated into BACTEC plus aerobic bottles. Inoculated bottles were incubated at 37 °C in a BACTEC 9050 automated analyser for up to 5 days and subcultured on fresh sheep blood agar, chocolate agar, and MacConkey agar when the machine indicated a positive signal. Organisms were identified by standard methods, including API20E and API20EN identification kits (Bio-Mérieux) or MALDI-TOF mass spectrometry (Bruker). Antimicrobial susceptibility testing of Kp was performed by VITEK automated machine and, when required, by minimum inhibitory concentrations (MICs) by E-test or the disk diffusion method using the guidelines established by the Clinical and Laboratory Standards Institute (CLSI). The study has been granted ethics approval from HTD's Ethic Committee (approval number: 1711/QĐ-BVBNĐ).

The rectal swabs/stools were sourced from two cohort studies. The first cohort involved 500 healthy children in Ho Chi Minh City, Vietnam, between 2014 and 2016 to longitudinally assess the incidence rate of dysentery[59]. Rectal swabs/stools from the final visit of each child were used for this study. Ethical approval for this study was granted by the Oxford University Tropical Research Ethics Committee (OxTREC No. 1058-13). The second cohort included healthy farmers in Dong Thap province (125 km WSW of HCMC) to assess the prevalence of AMR carriage before and after an intervention consisting of removing antimicrobial use in chicken farms[60]. A total of 350 rectal swabs were collected from farm households during 2020. All the samples originated from healthy adults. The study received ethics approval from the Oxford University Tropical Research Ethics Committee (OxTREC No. 503-20).

### Identification of hypervirulent *K. pneumoniae* isolates

To identify hvKp isolates from 700 Kp BSI isolates, a colony multiplex PCR assay with melting curve analysis was developed to target four virulence genes: *iucA*, *iroB*, *rmpA*, and *rmpA2*. Each target was associated with a specific melting temperature (Fig. S2 and Table S1). PCR amplifications were run on a LightCycler 480II (Roche applied sciences, UK) with the following thermal conditions: initial denaturation at 95 °C for 5 min, followed by 40 cycles of denaturation, 95 °C, for 10 s; annealing, 60 °C for 30 s; extension, 72 °C for 30 s. After PCR amplification, the melting curve analysis (MCA) was conducted in the same thermocycler at 65 °C to 95 °C and cooling cycle at 37 °C for 30 s.

Fluorescence was continuously measured, and the melting temperature (Tm) was calculated by plotting the negative derivative of fluorescence over temperature versus temperature (−dF/dT versus T). All Kp isolates harboring the *iucA* gene with/without other virulence genes were classified as hvKp[20]. For rectal swabs/stools, samples were initially plated on MacConkey agar containing 3 mg/L potassium tellurite to distinguish hypervirulent *Klebsiella* colonies from other bacteria. *K. pneumoniae* colonies were later confirmed by MALDI-TOF and underwent colony multiplex PCR for the detection of hvKp isolates. All the hvKp isolates were subjected to the string test and WGS.

### Whole genome sequencing

Genomic DNA was extracted using the Wizard Genomic DNA Extraction Kit (Promega, US) following the manufacturer's protocol. The Nextera XT Library Preparation Kit was used to prepare DNA sequencing libraries. WGS was performed on an Illumina MiSeq platform to generate 250 bp paired-end reads.

Plasmid DNA extraction was performed using a QIAGEN Plasmid MIDI kit (Qiagen) following the manufacturer's instructions. Nanopore sequencing libraries were generated from 200 ng of plasmid DNA using the SQK-RBK004 Rapid Barcoding Kit and then sequenced using the FLOW-FLG001 flow cell R9.4.1 (Oxford Nanopore Technologies). Nanopore sequencing was performed using MinKNOW software v1.13 to produce raw sequences in Fast5 format, which were subsequently subjected to base calling and de-multiplexing with Guppy v. 4.0[61].

### Genomic analysis

Raw Illumina reads were subjected to quality control using FastQC v0.11.5[62]. De novo assembly of Illumina reads, and hybrid assembly of Illumina and Nanopore reads were performed using Unicycler v.0.4.8b with default parameters[63]. Assembled sequences were annotated using Bakta v.1.0.4[64]. Kleborate v2.2.0[65] was used to determine the multilocus sequence types (MLSTs) with BIGSdb database[66], acquired AMR genes with CARD database[67], virulence genes with VFDB database[68], and K capsular types and O antigens with Kaptive database v.2.0.0[69]. SRST2 v0.2.0[70] was used to identify plasmid replicon types with PlasmidFinder database[71] and Abricate v1.0.0 (https://github.com/tseemann/abricate) was used to identify the plasmid origin of transfer site (*oriT*) using *oriT*DB database developed by Li and

collaborators[72]. For plasmids that failed to identify oriT within the *oriT*DB database, we searched for putative uncharacterized oriT regions. Briefly, oriTfinder (http://bioinfo-mml.sjtu.edu.cn/oriTfinder) was used to identify relaxase-encoding regions within these plasmids. Subsequently, RepeatFinder in Geneious v2022.1.1 (https://www.geneious.com/plugins/repeat-finder/) was used to identify the putative inverted repeats (IRs) neighboring the relaxase-encoding region, and MEME-MAST[73] was used to identify the conserved nick regions. The comparative genomics of *iuc3*-carrying plasmids were performed using Blast Ring Image Generator (BRIG)[74].

### SNP calling and phylogenetic analysis

Raw Illumina reads of all hvKp isolates were mapped against the reference genome of hvKp strain NTUH-K2044 (accession number: AP006725.1) using the RedDog pipeline v1.10b (https://github.com/katholt/RedDog). Briefly, RedDog used Bowtie2 v2.2.3 to map all raw reads to the reference sequence, and high-quality SNPs with Phred quality score ≥30 were extracted using SAMtools v1.1.3.1[75]. SNPs were subsequently filtered to remove those with fewer than five supporting reads or with >2.5 times the mean read depth (representing putative repeated sequences), or with ambiguous base calls. Core SNPs found in greater than 95% of isolates were extracted, resulting in a final alignment of 109,991 SNPs. Phylogenetic reconstruction was performed with IQ-TREE v.2.0[76], using the maximum likelihood (ML) method and the best-fit model GTR + ASC + G4, as determined by the software. Support for the ML tree was assessed through 1000 bootstrap replicates. The pairwise genetic distances (measured as a difference in the number of SNPs) between acute and carriage isolates within the same sequence type (STs) were calculated from the above SNP alignment using ape v4.1 and adegenet v2.0.1 packages in R v3.3.2.

To examine the phylogeny of ST23 hvKp isolates on a global scale, we integrated 59 ST23 genomes obtained from this study with an additional 176 ST23 genomes sourced from blood isolates available at the time of access from the Enterobase (https://enterobase.warwick.ac.uk/) (Supplementary Data 1). Raw reads were downloaded from the Enterobase using the ENAdatabase-Downloader script (https://github.com/chauvinhtth13/ENAdatabase-Downloader). Mapping analysis and SNP calling were performed as described above, using the same reference genome of hvKp strain NTUH-K2044 (ST23) and the RedDog pipeline v.1b.11. The snpTable2GenomeAlignment.py script from RedDog was employed to infer a pseudo-genome alignment, which was subsequently subjected to Gubbins v.2.34 to identify recombinant regions[77]. SNP-sites were used to extract SNPs from the recombination-free multi-FASTA alignment, resulting in an alignment of 8201 SNPs[78]. IQ-TREE was employed to reconstruct an ML tree from the SNP alignment with the best-fit model TVM + F + ASC + G4 and bootstrap analysis with 1000 replicates.

To explore the transmission dynamics of the virulence plasmid IncFIBK, we reconstructed a mapping-based plasmid phylogeny and assessed its congruence with the chromosomal phylogeny. Plasmid phylogenetic analysis employed the same mapping and SNP calling methods as described for the chromosomal phylogeny. The virulence plasmid pLVPK (accession number AY378100.1) served as the reference plasmid genome for this analysis. The final alignment, comprising 1568 SNPs, was utilized to construct a phylogeny using IQ-TREE, employing the best-fit model TVM + G4.

### Plasmid profiling

Crude plasmid extraction was performed for all hvKp isolates, using a modified Kado and Liu method[79]. The resulting plasmid DNA was subjected to electrophoresis in 0.7% agarose gel at 90 Voltage for 3 h, stained with Nancy-520 (2.5 mg/mL), and photographed. *E. coli* strain 39R861 carrying three plasmids with known sizes (36 kb, 63 kb, and 147 kb) was used as a marker[80].

### Transferability of the virulence plasmid

To investigate the transferability of the virulence plasmid, a conjugation experiment was performed between each of the 33 ciprofloxacin-susceptible hvKp strains (donor) and a ciprofloxacin-resistant clinical Kp ST15 strain (recipient). The selection of 33 donors aimed to encompass various sequence types (STs), both with and without the presence of additional non-virulence plasmids. Among 33 donors, 30 strains were resistant to tellurite (MIC >32 mg/L) and three were susceptible to tellurite (MIC <1 mg/L) but resistant to tetracycline (MIC >32 mg/L). The recipient Kp strain was identified from a patient with bloodstream infection in Vietnam, carrying a small Col plasmid of 2328 bp and a non-AMR phagemid of 109,938 bp. This strain exhibited chromosomal ciprofloxacin resistance (MIC = 16 mg/L), intermediate tetracycline resistance (MIC = 8 mg/L), and susceptibility to tellurite (MIC <1 mg/L). The donor and recipient were combined in a 1:1 ratio in LB broth and conjugated at 37 °C for 18 h. The transconjugants were selected on Mueller Hinton agar supplemented with ciprofloxacin (6 mg/L) plus tellurite (8 mg/L) or ciprofloxacin (6 mg/L) plus tetracycline (16 mg/L). Successful transfer of the virulence plasmid was verified by comparing the plasmid profiles of the donor, recipient, and transconjugants. Furthermore, the presence of plasmid-mediated virulence genes in the transconjugants was confirmed by qPCR.

### Reporting summary

Further information on research design is available in the Nature Portfolio Reporting Summary linked to this article.

## Data availability

All data used in this work are publicly available in the European Nucleotide Archive (ENA) under project numbers PRJEB64408 and PRJEB26814 (https://www.ebi.ac.uk/ena/browser/view/ PRJEB26814)[37]. Source data are provided with the paper. Source data are provided with this paper.

## Code availability

All software utilized for data analysis is open source.

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

## Acknowledgements

We would like to thank the healthcare staff at the Hospital for Tropical Diseases, Ho Chi Minh City, Vietnam, for their assistance in data collection for this study. The research was funded by the Wellcome International Training Fellowship (grant number: 222983/Z/21/Z) awarded to P.T.D and John Fell Oxford University Press Research Fund (reference: 0010734) awarded to M.C., M.A.R., and P.T.D. The funders had no role in study design, data collection and analysis, decision to publish, or preparation of the manuscript.

## Author contributions

Q.N. and N.T.P.Y. conducted experiments, performed data analysis and interpreted the results under the scientific guidance from P.T.D. Q.N. and N.T.P.Y. drafted and edited the paper and P.T.D. contributed in structuring and editing of the paper. P.T.D. provided oversight of the project. H.T.T., T.T.N.D., V.V.P., C.V. and N.T.N.T. conducted experiments and edited the paper. T.V.T., N.L.N.P., L.T.Q.N., N.P.H.L. and J.J.C. contributed to data acquisition in the field, and editing of the paper. S.B., G.T., M.A.R., M.C. and C.T.H. contributed to the data interpretation and editing of the paper. All authors read and approved the final draft.

## Competing interests

The authors declare no competing interests.

## Additional information

[1]Oxford University Clinical Research Unit, Ho Chi Minh City, Vietnam. [2]Hospital for Tropical Diseases, Ho Chi Minh City, Vietnam. [3]Centre for Tropical Medicine and Global Health, Nuffield Department of Medicine, University of Oxford, Oxford, UK. [4]Cambridge Institute of Therapeutic Immunology & Infectious Disease (CITIID) Department of Medicine, University of Cambridge, Cambridge, UK. [5]These authors contributed equally: Quynh Nguyen, Yen Thi Phuong Nguyen. ✉e-mail: duypt@oucru.org

