## [Peer Review File · Nature Communications]

Genomic insights unveil the plasmid transfer mechanism and epidemiology of hypervirulent *Klebsiella pneumoniae* in VietnamREVIEWER COMMENTS

Reviewer #1 (Remarks to the Author):

In their manuscript, Nguyen et al. investigate the prevalence, population structure, and virulence, AMR and plasmid features of hypervirulent *Klebsiella* from three collection of isolates (BSI, healthy adults and healthy children), and provide an interesting and detailed overview of the genomes sequenced from 'hvKp' in their collections - I have only a few remarks. The inclusion of genomes from healthy adult carriers adds to their insights, particularly as convergent AMR-virulence features were also detected in these isolates, but the authors should perhaps scale back the emphasis placed on the study encompassing infection vs healthy carriers/reported prevalence given that these collections don't span the same time scales i.e. 14.3% across a 10 year period of BSIs vs 13 isolates collected from farmers in 1 year, and also vary in screening sample sizes. Somewhat related to this, are the authors able to separate out BSI isolates collected from adults vs children, and is there a difference between the prevalence of Kp and hvKp between adult vs children? In the introduction, the authors do a great job of explaining that hvKp 'infections' generally lack a consensus definition, and that the true hypervirulent infections (liver abscess etc) are usually caused by distinct 'hvKp isolates' with have genomic/genotypic features = markers for these isolates. It's probably worth reminding the audience again in the discussion that the study collection encompass isolates that have the features typical of hvKp (*iucA*, ST, virulence plasmid) as opposed to being defined based on disease/true hypervirulent infection.

Other comments:

Lines 94-97: Mechanisms responsible for hvKp is more nuanced than capsule overproduction and more insights on this has been published in recent years (e.g. refer to Walker et al. papers PMID 32963003 and 30914502). Further, these papers have also revealed that *rmpA* belongs to a locus comprising *rmpC* and *rmpD*.

Line 247: By 8.8%, do the authors mean positive for Kp or any microorganism?

Line 248: maybe reiterate here that hvKp were identified/defined based on presence of the *iucA* PCR.

Lines 251-252: The stats here capture isolates that were *iucA*+ by PCR, which the authors use to define hvKp; can the authors clarify here if there were any isolates that were positive for *iro/rmpA* and/or *rmpA2* but not *iucA*? Similarly, what were the rates of Kp carriage (non-hvKp isolates) in the healthy adult and children cohorts?

Line 255: which subspecies? Perhaps the authors can also detail here (somewhere in this paragraph) that the two Kq isolates belonged to different STs.

Lines 259-261: I'm not sure how strong the stats would be (and therefore the significance of these statements are) given that there were so few carrier isolates vs BSIs..? Perhaps the authors can instead detail the ST breakdown of the carrier cohort by numbers and highlight that the distribution of STs look different compared to that of the BSI cohort?

Lines 263-264: Are these K loci numbers driven by STs? i.e. K1 in ST23? It would be helpful to include numbers of ST-KL combinations like what was done for the O antigens in lineages 265-266.

Lines 281-285: Do these four clusters correspond with those that have been defined in previous CG23 studies. For example, a few CG23 lineages have been described in Lam and Wyres et al. 2018 (<https://pubmed.ncbi.nlm.nih.gov/30006589/>). Can the authors also state how many of their study isolates belong to their four defined clusters?

Lines 287-288: Is there a reason why the authors use only a single gene in reference to the *iuc* and *iro* locus? Similarly, *rmpA* is part of locus.

Line 297: The authors have used the *iroX-iucX-clb-X* etc. to label the virulence loci and their lineages within their collection, with no separation between plasmid vs. chromosomal. This is fine but the labelling of '*iro3-iuc3*' here may imply that these are co-located on the same plasmid (e.g. like *iuc1+iro1*) when *iro3* and *iuc3* are located on different mobile elements (ICEKp1 vs plasmid).

Line 308: Did the authors look into how these AMR genes were integrated into the chromosomes?

Lines 308-309: Do these 5 isolates carry the same plasmid?

Lines 331-332: Despite the labelling of these virulence loci as lineage 3, it has been previously established that *iuc3* is plasmid-encoded while *iro3* and *rmp3* is ICEKp1-encoded.

Lines 346-347: How many of these also have *iro1*, *rmp1* and *rmpA2*?

Lines 375-377: Is it possible to quantify the rates at which each of these different plasmid transfer events occur?

Lines 460-461: There also shouldn't be a reliance/emphasis placed on the string test for identifying hvKp as the string test can also be affected by culturing variables.

Figure 1: Font size of the column headings, tip labels (have the authors considered displaying this information in the columns?) and probably the legend are far too small. Should probably also define the abbreviations for the virulence and AMR loci in the figure text description.

Figure 3: How are the authors defining 'plasmid phylogenetic clusters'? The third cluster (i.e. blue shading) shows a deeper split between the purple and the red? Similarly, the labels in this figure are far too small and can probably be visually represented instead.

Table 1: Why are iucA/iroB being used as markers for an entire locus when the authors have verified the presence of the entire locus in the WGS data?

Figure S3 Line 40: replace 'different alleles of virulence genes' with 'different lineages of virulence loci'

Figure S4: Would be helpful to label the other iuc3 plasmids with the ST of the host strain? Labels of the loci are too small and the font colour of iuc locus difficult to read.

Reviewer #2 (Remarks to the Author):

This is the second revision of this manuscript (which I previously reviewed for a different journal). In the first iteration I flagged a few issues with the Bayesian phylogenetic analyses, which the authors have now removed. I raised no further concerns.

Reviewer #3 (Remarks to the Author):

This study collected blood culture specimens from a hospital over a period of ten years, rectal swabs from healthy farmers in another province in 2020, and rectal swabs from children in Ho Chi Minh City over three consecutive years, aiming to identify HVKP. Although it costs a lot of efforts, the sampling criteria based on different time and location lack scientific rationale, resulting in incomparable results. Furthermore, these data cannot represent the entire country of Vietnam, the title of this article needs to be reconsidered. Overall, I don't recommend this manuscript to be published in the journal for current stage. However, I list my comments below so the author could refer to them while revising the manuscript before re-submit it to other journals.

1: To conduct a comparative study between children and adults, it would be more appropriate to classify and study the 700 cases of KP infection from the hospital according to age. As less HVKP was isolated from rectal swabs of healthy adults and children, the description in this regard can be considered for removal or downplaying.

2. It is recommended to download publicly available data from more public databases to enrich your phylogenetic tree analysis to include HVKP data from healthy individuals. Constructing a phylogenetic tree of various ST types together, as shown in Figure 1, makes little sense, because the results are likely to be clusters of the same ST type. A phylogenetic analysis focusing on the same ST would be more meaningful, as shown in Figure S3.

3. It is unclear from the description whether the phylogenetic analysis of strains differentiated between chromosomal and plasmid DNA. It is advised to focus on the chromosomal genome when analyzing the strains, excluding the interference from plasmids.

4. I am unsure about the expression of plasmid types in the text. For instance, in line 299, what

does "+/-" mean in "IncFIBK+/-IncHI1B"?

5. Could you please explain the reason for analysis of iuc3-carrying virulence plasmids? It is not the majority virulence plasmid in the study of HVKP in this article.

6. The result description in the article makes me confused. After describing the genetic structure of iuc3 plasmid, the study performed experiments to investigate the iuc1 plasmid. Is it possible to analyze the genetic structure and transfer mechanism of the iuc1 plasmid in depth, instead of focusing on iuc3 plasmid?

7. The results and legend of Figure 4 are somewhat difficult to comprehend. Why are there newly appeared plasmids (blue arrows)? Could it be contamination from other plasmid-carrying strains or changes in plasmid size after transfer? It requires re-experiment and further experimental validation or sequencing to provide clear explanations. Additionally, the strip positions of the green and white arrows are the same, but do they represent plasmids of different sizes?

8. Please provide a summary of the highlights or significant discoveries in the abstract.

Re: Submission to Nature Communications

Dear Editor,

I wish the modified manuscript entitled “**Genomic insights unveil the plasmid transfer mechanism and epidemiology of hypervirulent *Klebsiella pneumoniae* in Vietnam**”, for which I am the corresponding author, to be considered for publication in your journal. We thank the editors and the reviewers for their helpful contributions and have endeavored to make the required modifications to the manuscript. Our responses are outlined below.

Reviewers comments

Reviewer #1 (Remarks to the Author):

In their manuscript, Nguyen et al. investigate the prevalence, population structure, and virulence, AMR and plasmid features of hypervirulent *Klebsiella* from three collection of isolates (BSI, healthy adults and healthy children), and provide an interesting and detailed overview of the genomes sequenced from 'hvKp' in their collections - I have only a few remarks. The inclusion of genomes from healthy adult carriers adds to their insights, particularly as convergent AMR-virulence features were also detected in these isolates, but the authors should perhaps scale back the emphasis placed on the study encompassing infection vs healthy carriers/reported prevalence given that these collections don't span the same time scales i.e. 14.3% across a 10 year period of BSIs vs 13 isolates collected from farmers in 1 year, and also vary in screening sample sizes. Somewhat related to this, are the authors able to separate out BSI isolates collected from adults vs children, and is there a difference between the prevalence of Kp and hvKp between adult vs children?

We appreciate the reviewer's positive feedback.

Our primary objective was to assess the diversity and overlap of hypervirulent *Klebsiella pneumoniae* (hvKp) genotypes in bloodstream infections (BSIs) and among healthy carriers. We acknowledge the limitations and did not aim to directly compare the prevalence of hvKp in these two groups. We have accordingly adjusted relevant statements in the manuscript to temper our conclusions.

All BSI isolates were obtained from adult patients aged over 25 years old. The absence of hvKp infections in children is supported by the observation that no hvKp isolates were found in the rectal swabs collected from healthy children. Furthermore, there have been very few reports of hvKp infections in children in the published literature.

Due to the primary focus of the study on hvKp and the significant workload involved in bacterial identification from the rectal swabs, our initial attention was not directed towards Kp. Consequently, while Kp was observed in the rectal swabs of healthy children, there was no recorded prevalence of Kp during the early stages of the study. However, in the later stages of the investigation, the prevalence of Kp in the rectal swabs of healthy adults was recorded at 12%. In summary, our study did not specifically compare the prevalence of Kp between adults and children, as it was not within our primary objective.

In the introduction, the authors do a great job of explaining that hvKp 'infections' generally lack a consensus definition, and that the true hypervirulent infections (liver abscess etc) are usually caused by distinct 'hvKp isolates' with have genomic/genotypic features = markers for these isolates. It's probably worth reminding the audience again in the discussion that the study collection encompass isolates that have the features typical of hvKp (*iucA*, *ST*, virulence plasmid) as opposed to being defined based on disease/true hypervirulent infection.

We thank the reviewer for your comment.

We have added details in the Discussion following your recommendation (line 457).

Other comments:

Lines 94-97: Mechanisms responsible for hvKp is more nuanced than capsule overproduction and more insights on this has been published in recent years (e.g. refer to Walker et al. papers PMID 32963003 and 30914502). Further, these papers have also revealed that *rmpA* belongs to a locus comprising *rmpC* and *rmpD*.

We thank the reviewer for your comment.

We have revised the sentence to prevent any misinterpretation regarding the association between the hypermuroid phenotype and capsule overproduction (line 111).

Line 247: By 8.8%, do the authors mean positive for Kp or any microorganism?

The positivity rate of 8.8% was observed for clinically relevant pathogen (bacteria/fungi) from blood cultures obtained from patients with clinical syndromes indicative of bloodstream infections.

Line 248: maybe reiterate here that hvKp were identified/defined based on presence of the *iucA* PCR.

This sentence has been modified following the reviewer's recommendation (line 304). It now reads:

"Out of the total 700 Kp isolates responsible for bloodstream infections (BSIs) at HTD between 2010 and 2020, 100 isolates (14.3%) were identified as hvKp based on the results of *iucA* PCR"

Lines 251-252: The stats here capture isolates that were *iucA*+ by PCR, which the authors use to define hvKp; can the authors clarify here if there were any isolates that were positive for *iro/rmpA*

and/or *rmpA2* but not *iucA*? Similarly, what were the rates of Kp carriage (non-hvKp isolates) in the healthy adult and children cohorts?

No isolates were positive for *iro/rmpA* and/or *rmpA2* but negative for *iucA*. If the isolates lacked *iucA*, they also lacked other virulence genes. As described above, the rate of Kp carriage in healthy adults was 12% while the prevalence of Kp in healthy children was not documented.

Line 255: which subspecies? Perhaps the authors can also detail here (somewhere in this paragraph) that the two Kq isolates belonged to different STs.

The subspecies has been given in the Results section (line 312). Their STs have been included in the Discussion section (lines 467-469).

Lines 259-261: I'm not sure how strong the stats would be (and therefore the significance of these statements are) given that there were so few carrier isolates vs BSIs..? Perhaps the authors can instead detail the ST breakdown of the carrier cohort by numbers and highlight that the distribution of STs look different compared to that of the BSI cohort?

We have removed the stats and revised these sentences to include the breakdown of STs by the number of isolates, and provided data on the distribution of major STs in BSIs and the healthy adults (lines 316-318).

Lines 263-264: Are these K loci numbers driven by STs? i.e. K1 in ST23? It would be helpful to include numbers of ST-KL combinations like what was done for the O antigens in lineages 265-266. The association between KL and ST has been described in the manuscript (lines 321-323).

Lines 281-285: Do these four clusters correspond with those that have been defined in previous CG23 studies. For example, a few CG23 lineages have been described in Lam and Wyres et al. 2018 (<https://pubmed.ncbi.nlm.nih.gov/30006589/>). Can the authors also state how many of their study isolates belong to their four defined clusters?

We have mapped our ST23 isolates onto the global phylogenetic structure described in Lam and Wyres et al. 2018. Our phylogenetic clusters C2, C3 and C4 are nested within the global sublineage CG23-1 (<https://pubmed.ncbi.nlm.nih.gov/30006589/>), whereas the C1 cluster belongs to the CG23 lineage.

Lines 287-288: Is there a reason why the authors use only a single gene in reference to the *iuc* and *iro* locus? Similarly, *rmpA* is part of locus.

From genomic analysis, we found that the all isolates carried full operons of *iuc* and *iro* genes. Here, we referred to a single gene to facilitate the comparison between the gene-based PCR assay and WGS. We have revised the sentence to avoid confusion in data interpretation (line 353). It now reads:

“Genomic analysis revealed that all isolates possessed at least two plasmid-encoded siderophores biosynthesis genes: *iuc* (aerobactin) and *iro* (salmochelins)”

Line 297: The authors have used the *iroX-iucX-clb-X* etc. to label the virulence loci and their lineages within their collection, with no separation between plasmid vs. chromosomal. This is fine but the labelling of '*iro3-iuc3*' here may imply that these are co-located on the same plasmid (e.g. like *iuc1+iro1*) when *iro3* and *iuc3* are located on different mobile elements (ICEKp1 vs plasmid).

We thank the reviewer for your comment.

We have revised the sentence accordingly (lines 362-364). The sentence now reads:

“Most hvKp strains (108/111) harbored the *iro1-iuc1* virulence profile, with the exception of three isolates carrying the plasmid-encoded *iuc3* and the ICEKp1-encoded (2 ST25) and IS100-encoded (1 ST828) chromosomal *iro3* gene.”

Line 308: Did the authors look into how these AMR genes were integrated into the chromosomes?
We have provided the information about the IS elements associated with chromosomal integration of these AMR genes in the manuscript. The sentence now reads:

“Two of them harbored an AMR gene on the chromosome, including one ST23 isolate (*ISEc9-bla_{CTX-M-14}*) and one ST86 isolate (*ISKpn74-sul1*).”

Lines 308-309: Do these 5 isolates carry the same plasmid?

These 5 isolates did not carry the same IncFII plasmid. In fact, their genetic structures differ according to Nanopore sequencing data. We have added further information in the manuscript (line 381)

Lines 331-332: Despite the labelling of these virulence loci as lineage 3, it has been previously established that *iuc3* is plasmid-encoded while *iro3* and *rmp3* is ICEKp1-encoded.

We thank the reviewer for the information. We have removed the sentence.

Lines 346-347: How many of these also have *iro1*, *rmp1* and *rmpA2*?

Among 108 *iuc1*-encoding IncFIBK+/-IncHIB virulence plasmids, 63 isolates carried *iro1* and *rmp1* while 36 harbored *iro1*, *rmp1* and *rmpA2*. Further details regarding the distribution of virulence factors (*rmp*, *ybt*, *clb*, *iuc*, *iro*) among hvKp isolates can be found in Figure 1.

Lines 375-377: Is it possible to quantify the rates at which each of these different plasmid transfer events occur?

Here, we did not aim to compare the conjugation frequency of each plasmid transfer event. Therefore, we did not count the number of donors and recipients. However, we observed variations in the number of transconjugants across different plasmid conjugation events after 18-hour conjugation. Specifically, the highest number of transconjugants was associated with the 96-kb IncFII helper plasmid (donor strain: D3) while the lowest was associated with the 122-kb IncFII:IncFIA:IncN helper plasmid (donor strain:D4) (Please see the Figure below). Although these observations suggest differing transfer frequencies, we did not quantify the rates.

Lines 460-461: There also shouldn't be a reliance/emphasis placed on the string test for identifying hvKp as the string test can also be affected by culturing variables.

We thank the reviewer for your comment.

While we acknowledge the potential impact of culturing conditions on the string test, the paragraph is primarily dedicated to discussing the genetic structure and vehicles of the *iuc3* virulence plasmid. Therefore, we feel that adding the information about string test may detract from the main message of the paragraph.

Alternatively, we have incorporated this information, along with the relevant citation, into the Introduction section of the manuscript where the context is more suitable (line 112).

Figure 1: Font size of the column headings, tip labels (have the authors considered displaying this information in the columns?) and probably the legend are far too small. Should probably also define the abbreviations for the virulence and AMR loci in the figure text description.

We have revised the Figure 1 to enhance visualization and data interpretation.

Figure 3: How are the authors defining 'plasmid phylogenetic clusters'? The third cluster (i.e. blue shading) shows a deeper split between the purple and the red? Similarly, the labels in this figure are far too small and can probably be visually represented instead.

The plasmid phylogenetic clusters were defined based on bootstrap support values as mentioned in the Results section. An error in the Figure text description has been corrected to reflect this information:

“The blue shaded boxes highlight sequence types (STs) that are associated with incongruences between the chromosomal and plasmid phylogenies.”

Furthermore, the label of Figure 3 has been modified to improve its visualization.

Table 1: Why are *iucA/iroB* being used as markers for an entire locus when the authors have verified the presence of the entire locus in the WGS data?

In Table 1, we utilized single markers to facilitate the comparison between PCR detection and WGS, although the entire locus information for each virulence factor is available. To ensure clarity and consistency with published literature, we have updated the details of the key virulence factors in Table 1.

Figure S3 Line 40: replace 'different alleles of virulence genes' with 'different lineages of virulence loci'

This text has been modified accordingly.

Figure S4: Would be helpful to label the other *iuc3* plasmids with the ST of the host strain? Labels of the loci are too small and the font colour of *iuc* locus difficult to read.

The label has been updated to improve visualization, and the ST of the host strain has been included for additional clarity.

Reviewer #2 (Remarks to the Author):

This is the second revision of this manuscript (which I previously reviewed for a different journal). In the first iteration I flagged a few issues with the Bayesian phylogenetic analyses, which the authors have now removed. I raised no further concerns.

We appreciate the reviewer for your thoughtful consideration.

Reviewer #3 (Remarks to the Author):

This study collected blood culture specimens from a hospital over a period of ten years, rectal swabs from healthy farmers in another province in 2020, and rectal swabs from children in Ho Chi Minh City over three consecutive years, aiming to identify HVKP. Although it costs a lot of efforts, the sampling criteria based on different time and location lack scientific rationale, resulting in incomparable results.

As mentioned above, our primary objective was not to directly compare the prevalence of hvKp in bloodstream infections and healthy cohorts. Instead, our focus was on assessing the diversity and overlap of hvKp genotypes in BSIs and among healthy carriers. Such information is scarce in low-resource settings like Vietnam. Throughout the manuscript, we have avoided making explicit prevalence comparisons. In response to Reviewer 1's feedback on a similar concern, we have removed any sentences that may have implied such comparisons. Furthermore, we have made additional revisions in the Discussion section to temper any conclusions in this regard.

Furthermore, these data cannot represent the entire country of Vietnam, the title of this article needs to be reconsidered. Overall, I don't recommend this manuscript to be published in the journal for current stage. However, I list my comments below so the author could refer to them while revising the manuscript before re-submit it to other journals.

We appreciate the reviewer's consideration regarding the representativeness of our data and the title of our manuscript. As with other genomic surveillance studies, the issue of data representativeness is often debated. While we acknowledge that our study may not capture the entire spectrum of hvKp infections across Vietnam, the title accurately reflects the specific focus and findings within our defined context. Our study provides valuable insights into hvKp genotypes in the studied region, contributing to the existing literature on this topic. However, we will ensure to clarify the scope and limitations of our study in the manuscript to avoid any misinterpretation.

1: To conduct a comparative study between children and adults, it would be more appropriate to classify and study the 700 cases of KP infection from the hospital according to age. As less HVKP was isolated from rectal swabs of healthy adults and children, the description in this regard can be considered for removal or downplaying.

Again, our primary focus in this study was not on KP infections or comparing children to adults. Rather, our research aimed to investigate hvKp and understand the diversity and overlap of hvKp genotypes in BSIs and among healthy carriers. It is important to note that while *K. pneumoniae* infections can occur in children (data not shown), all hvKp isolates from BSIs in our study were obtained from adults aged over 25 years old. The absence of hvKp infections in children is supported by the observation that no hvKp isolates were found in the rectal swabs obtained from healthy children. This finding aligns with existing literature, which has reported very few cases of hvKp infections in children. Our study findings contribute to understanding the reasons behind the low prevalence of hvKp infections occurring in children.

2. It is recommended to download publicly available data from more public databases to enrich your phylogenetic tree analysis to include HVKP data from healthy individuals. Constructing a phylogenetic tree of various ST types together, as shown in Figure 1, makes little sense, because the results are likely to be clusters of the same ST type. A phylogenetic analysis focusing on the same ST would be more meaningful, as shown in Figure S3.

We thank the reviewer's suggestion regarding the inclusion of additional hvKp isolates from healthy individuals to strengthen our conclusions. However, it's important to note that there are currently no publicly available data from within Vietnam for such inclusion. We surmise that it is essential for infections and carriage data to originate from the same population and setting to draw meaningful conclusions.

The phylogenetic trees presented in Figure 1 and Figure S3 serve distinct purposes within our study. Figure 1 showcases the clustering of all sequence types (STs), providing an overview of their distribution of virulence factors, acquired antimicrobial resistance (AMR) genes, and plasmid replicon types. This comprehensive view allows us to discern similarities and differences in key phenotypic and genotypic features both within and between STs. Additionally, the topology of this tree is compared with that of the plasmid tree to understand the potential for co-evolution and horizontal transfer of the virulence plasmid.

In contrast, the phylogenetic tree in Figure S3 specifically focuses on delineating the transmission dynamics of the ST23 population, a predominant clone circulating in Vietnam. Here, we highlight its transmission patterns, driven by multiple introductions and endemic expansion, providing insights into the spread of this particular lineage within the population.

3. It is unclear from the description whether the phylogenetic analysis of strains differentiated between chromosomal and plasmid DNA. It is advised to focus on the chromosomal genome when analyzing the strains, excluding the interference from plasmids.

When analyzing the strains, we did focus exclusively on the chromosomal genome as outlined in the Methods section (line 228). We mapped our raw read data against the reference genome of hvKp strain NTUH -K2044 (accession number: AP006725), which contains solely the chromosomal genome. Furthermore, for the plasmid phylogeny analysis, we utilized only the virulence plasmid pLVPK (accession number: AY378100) as the reference genome (Line 262).

4. I am unsure about the expression of plasmid types in the text. For instance, in line 299, what does "+/-" mean in "IncFIBK+/-IncHI1B"?

The virulence plasmid can exist as either a single replicon IncFIBK plasmid or a multi-replicon IncFIBK-IncHIB plasmid, as documented in previous publications (PMID: 33970792 and PMID: 30371343). We denote this as IncFIBK+/-IncHI1B, signifying that the IncFIBK replicon is consistently present on the virulence plasmid, while the presence of IncHI1B may vary among strains.

5. Could you please explain the reason for analysis of iuc3-carrying virulence plasmids? It is not the majority virulence plasmid in the study of HVKP in this article.

While iuc1 virulence plasmids are more prevalent in our study, iuc3 virulence plasmids exhibit epidemic potential due to their distinct genetic profile and self-transmissible nature. These iuc3 plasmids carry both virulence and conjugation modules, along with antimicrobial resistance genes, enabling them to spread in environments with high antibiotic selection pressure. Importantly, iuc3-carrying hvKp strains may not exhibit the hypermuroid phenotype, typically detected by traditional microbiology methods like the string test. This highlights the need for clinical microbiologists to be aware of alternative detection methods for iuc3-carrying strains. Furthermore, the majority of iuc3 plasmids in hvKp are believed to originate from animal sources, suggesting a zoonotic reservoir that warrants further investigation on a broader scale, encompassing both hosts and transmission mechanisms.

6. The result description in the article makes me confused. After describing the genetic structure of iuc3 plasmid, the study performed experiments to investigate the iuc1 plasmid. Is it possible to analyze the genetic structure and transfer mechanism of the iuc1 plasmid in depth, instead of focusing on iuc3 plasmid?

The genetic structure of iuc1 virulence plasmids has been extensively studied (PMID: 15276215 and PMID: 29338592), whereas there have been limited reports on the genetic structures of iuc3 virulence plasmids. We conducted experiments and described conjugation results for both iuc1 and iuc3 plasmids as outlined in the Results section (line 433). Given that iuc3 plasmids are self-transmissible, the transfer mechanism was specifically examined for the non-conjugative iuc1 plasmids.

7. The results and legend of Figure 4 are somewhat difficult to comprehend. Why are there newly appeared plasmids (blue arrows)? Could it be contamination from other plasmid-carrying strains or changes in plasmid size after transfer? It requires re-experiment and further experimental validation or sequencing to provide clear explanations. Additionally, the strip positions of the green and white arrows are the same, but do they represent plasmids of different sizes?

We have revised the text and legend of Figure 4 to enhance clarity and facilitate data interpretation. The newly appearing blue arrows denote plasmids with sizes differing from those in the donor strains. Contamination is highly unlikely as the plasmid-carrying transconjugants must survive in selective media containing both tellurite and antibiotics. It is probable that these new plasmid bands are resulting from plasmid co-integration or recombination events, as demonstrated in previous studies (PMID: 34294113, PMID: 35967858 and PMID: 34075192).

In each conjugation experiment, we randomly selected 5 colonies to compare the number and size of plasmid bands between donors, recipient, and transconjugants. It is evident that such plasmid co-integration or recombination events do not occur in all transfer events, a finding consistent with previous studies. Each conjugation experiment was conducted at least twice to confirm the transferability of the virulence plasmids, and the newly emerged bands were also observed in other sets of experiments (data not shown).

The white arrows correspond to a phagemid present in recipient and transconjugants, while the green arrows represent distinct phagemids found in the donors. Their strip positions appear similar because they are very similar in sizes (white=109,938 bp; green=109,376 bp/D2 strain, and 106,291 bp/D1 strain).

8. Please provide a summary of the highlights or significant discoveries in the abstract.
We have revised the abstract to highlight our significant discoveries.

Yours sincerely,

Dr Pham Thanh Duy, PhD

Wellcome International Training Fellow
Head of Molecular Epidemiology Group
Oxford University Clinical Research Unit
764 Vo Van Kiet, District 5, Ho Chi Minh, Vietnam

Email: duypt@oucru.org

Profile: https://www.researchgate.net/profile/Duy_Pt

ORCID: <https://orcid.org/0000-0001-7029-9210>

REVIEWERS' COMMENTS

Reviewer #1 (Remarks to the Author):

The authors have done a great job of adequately addressing reviewer concerns raised in the first round of review. I only have one minor comment remaining:

Line 365: the phrasing 'IS100-encoded' is a bit odd, do the authors mean 'flanked'? Further, *iro3* is a locus and not a gene.

Reviewer #3 (Remarks to the Author):

The revised manuscript has been improved, and the authors have addressed some concerns. However, the data sampling strategy is unsatisfactory, and the genome analysis lacks valuable findings. It is possible that there is a lack of epidemiological data in Vietnam, but this manuscript may be better suited for other journals that prioritize the publication of surveillance data.

Additional comments in the attached file.

Reviewer #3 (Attachment):

This study collected blood culture specimens from a hospital over a period of ten years, rectal swabs from healthy farmers in another province in 2020, and rectal swabs from children in Ho Chi Minh City over three consecutive years, aiming to identify HVKP. Although it costs a lot of efforts, the sampling criteria based on different time and location lack scientific rationale, resulting in incomparable results.

As mentioned above, our primary objective was not to directly compare the prevalence of hvKp in bloodstream infections and healthy cohorts. Instead, our focus was on assessing the diversity and overlap of hvKp genotypes in BSIs and among healthy carriers. Such information is scarce in low-resource settings like Vietnam. Throughout the manuscript, we have avoided making explicit prevalence comparisons. In response to Reviewer 1's feedback on a similar concern, we have removed any sentences that may have implied such comparisons. Furthermore, we have made additional revisions in the Discussion section to temper any conclusions in this regard.

Furthermore, these data cannot represent the entire country of Vietnam, the title of this article needs to be reconsidered. Overall, I don't recommend this manuscript to be published in the journal for current stage. However, I list my comments below so the author could refer to them while revising the manuscript before re-submit it to other journals.

We appreciate the reviewer's consideration regarding the representativeness of our data and the title of our manuscript. As with other genomic surveillance studies, the issue of data representativeness is often debated. While we acknowledge that our study may not capture the entire spectrum of hvKp infections across Vietnam, the title accurately reflects the specific focus and findings within our defined context. Our study provides valuable insights into hvKp genotypes in the studied region, contributing to the existing literature on this topic. However, we will ensure to clarify the scope and limitations of our study in the manuscript to avoid any misinterpretation.

I agree it is necessary to screen for HVKP in healthy individuals and conduct such investigations in Vietnam. However, I do not believe that the study can adequately reflect the entire situation in Vietnam, and combining three entirely unrelated cohorts seems unnecessary. As you stated, 'We surmise that it is essential for infections and carriage data to originate from the same population and setting to draw meaningful conclusions.' If possible, I would propose undertaking a nationwide scientifically based sampling survey of healthy individuals. As for the current study, considering the limited number of HVKP cases from healthy individuals, it might be advisable to remove the healthy cohort and focus solely on the study of HVKP in bloodstream infections from one hospital in Vietnam over the past decade. This adjustment would help reduce unnecessary complexity and focus the research more effectively.

1. To conduct a comparative study between children and adults, it would be more appropriate to classify and study the 700 cases of KP infection from the hospital according to age. As less HVKP was isolated from rectal swabs of healthy adults and children, the description in this regard can be considered for removal or downplaying.

Again, our primary focus in this study was not on KP infections or comparing children to adults. Rather, our research aimed to investigate hvKp and understand the diversity and overlap of hvKp genotypes in BSIs and among healthy carriers. It is important to note that while *K. pneumoniae* infections can occur in children (data not shown), all hvKp isolates from BSIs in our study were obtained from adults aged over 25 years old. The absence of hvKp infections in children is supported

by the observation that no hvKp isolates were found in the rectal swabs obtained from healthy children. This finding aligns with existing literature, which has reported very few cases of hvKp infections in children. Our study findings contribute to understanding the reasons behind the low prevalence of hvKp infections occurring in children.

I can understand the desire to use of all the available data, but that doesn't mean we should simply lump everything together like a mishmash. If the goal is to report on the absence of HVKP in healthy children, why not write a separate dedicated article for that study?

2. It is recommended to download publicly available data from more public databases to enrich your phylogenetic tree analysis to include HVKP data from healthy individuals. Constructing a phylogenetic tree of various ST types together, as shown in Figure 1, makes little sense, because the results are likely to be clusters of the same ST type. A phylogenetic analysis focusing on the same ST would be more meaningful, as shown in Figure S3.

We thank the reviewer's suggestion regarding the inclusion of additional hvKp isolates from healthy individuals to strengthen our conclusions. However, it's important to note that there are currently no publicly available data from within Vietnam for such inclusion. We surmise that it is essential for infections and carriage data to originate from the same population and setting to draw meaningful conclusions.

The phylogenetic trees presented in Figure 1 and Figure S3 serve distinct purposes within our study. Figure 1 showcases the clustering of all sequence types (STs), providing an overview of their distribution of virulence factors, acquired antimicrobial resistance (AMR) genes, and plasmid replicon types. This comprehensive view allows us to discern similarities and differences in key phenotypic and genotypic features both within and between STs. Additionally, the topology of this tree is compared with that of the plasmid tree to understand the potential for co-evolution and horizontal transfer of the virulence plasmid.

In contrast, the phylogenetic tree in Figure S3 specifically focuses on delineating the transmission dynamics of the ST23 population, a predominant clone circulating in Vietnam. Here, we highlight its transmission patterns, driven by multiple introductions and endemic expansion, providing insights into the spread of this particular lineage within the population.

Thanks for your explanation. Figure S3 seems to be more valuable than Figure 1, and I suggest moving Figure 1 to the supplementary files.

3. It is unclear from the description whether the phylogenetic analysis of strains differentiated between chromosomal and plasmid DNA. It is advised to focus on the chromosomal genome when analyzing the strains, excluding the interference from plasmids.

When analyzing the strains, we did focus exclusively on the chromosomal genome as outlined in the Methods section (line 228). We mapped our raw read data against the reference genome of hvKp strain NTUH -K2044 (accession number: AP006725), which contains solely the chromosomal genome. Furthermore, for the plasmid phylogeny analysis, we utilized only the virulence plasmid pLVPK (accession number: AY378100) as the reference genome (Line 262).

Thanks for clarification. Did you conduct data cleaning before mapping with the reads?

4. I am unsure about the expression of plasmid types in the text. For instance, in line 299, what does "+/-" mean in "IncFIBK+/-IncHI1B"?

The virulence plasmid can exist as either a single replicon IncFIBK plasmid or a multi-replicon IncFIBK-IncHIB plasmid, as documented in previous publications (PMID: 33970792 and PMID: 30371343). We denote this as IncFIBK+/-IncHI1B, signifying that the IncFIBK replicon is consistently present on the virulence plasmid, while the presence of IncHI1B may vary among strains.

Thanks for clarification.

5. Could you please explain the reason for analysis of *iuc3*-carrying virulence plasmids? It is not the majority virulence plasmid in the study of HVKP in this article.

While *iuc1* virulence plasmids are more prevalent in our study, *iuc3* virulence plasmids exhibit epidemic potential due to their distinct genetic profile and self-transmissible nature. These *iuc3* plasmids carry both virulence and conjugation modules, along with antimicrobial resistance genes, enabling them to spread in environments with high antibiotic selection pressure. Importantly, *iuc3*-carrying hvKp strains may not exhibit the hypermucooid phenotype, typically detected by traditional microbiology methods like the string test. This highlights the need for clinical microbiologists to be aware of alternative detection methods for *iuc3*-carrying strains. Furthermore, the majority of *iuc3* plasmids in hvKp are believed to originate from animal sources, suggesting a zoonotic reservoir that warrants further investigation on a broader scale, encompassing both hosts and transmission mechanisms.

Thank you for the explanation. When analyzing *iuc3*, it would be more convincing to find additional data in public databases to analyze together?

6. The result description in the article makes me confused. After describing the genetic structure of *iuc3* plasmid, the study performed experiments to investigate the *iuc1* plasmid. Is it possible to analyze the genetic structure and transfer mechanism of the *iuc1* plasmid in depth, instead of focusing on *iuc3* plasmid?

The genetic structure of *iuc1* virulence plasmids has been extensively studied (PMID: 15276215 and PMID: 29338592), whereas there have been limited reports on the genetic structures of *iuc3* virulence plasmids. We conducted experiments and described conjugation results for both *iuc1* and *iuc3* plasmids as outlined in the Results section (line 433). Given that *iuc3* plasmids are self-transmissible, the transfer mechanism was specifically examined for the non-conjugative *iuc1* plasmids.

Thank you for the explanation. Although I feel that the research on *iuc3* have little value, as the manuscript does not demonstrate an increasing trend of *iuc3* within hvkp.

7. The results and legend of Figure 4 are somewhat difficult to comprehend. Why are there newly appeared plasmids (blue arrows)? Could it be contamination from other plasmid-carrying strains or changes in plasmid size after transfer? It requires re-experiment and further experimental validation or sequencing to provide clear explanations. Additionally, the strip positions of the green and white arrows are the same, but do they represent plasmids of different sizes?

We have revised the text and legend of Figure 4 to enhance clarity and facilitate data interpretation. The newly appearing blue arrows denote plasmids with sizes differing from those in the donor strains. Contamination is highly unlikely as the plasmid-carrying transconjugants must survive in selective media containing both tellurite and antibiotics. It is probable that these new plasmid bands are resulting from plasmid co-integration or recombination events, as demonstrated in previous studies (PMID: 34294113, PMID: 35967858 and PMID: 34075192).

In each conjugation experiment, we randomly selected 5 colonies to compare the number and size of plasmid bands between donors, recipient, and transconjugants. It is evident that such plasmid co-integration or recombination events do not occur in all transfer events, a finding consistent with previous studies. Each conjugation experiment was conducted at least twice to confirm the transferability of the virulence plasmids, and the newly emerged bands were also observed in other sets of experiments (data not shown).

The white arrows correspond to a phagemid present in recipient and transconjugants, while the green arrows represent distinct phagemids found in the donors. Their strip positions appear similar because they are very similar in sizes (white=109,938 bp; green=109,376 bp/D2 strain, and 106,291 bp/D1 strain).

Thank you for the explanation. I suggest the plasmid profiling is best done with S1-PFGE and southern blotting.

8. Please provide a summary of the highlights or significant discoveries in the abstract.

We have revised the abstract to highlight our significant discoveries.

While there have been articles documenting the significance of *nic* sites in plasmid transfer, it is essential to note that the sites identified in this study did not precisely match those previously reported. The distribution patterns of *nic* sites and plasmids observed in clinical strains do not definitively exclude the influence of other factors. To establish the functions of these sites, validation through mutants is necessary.

It is noteworthy that a recent study (PMID: 38042959) reported that *K. pneumoniae* carrying virulence genes may not necessarily exhibit high virulence, so defining hvKP should be more cautious.

This study collected blood culture specimens from a hospital over a period of ten years, rectal swabs from healthy farmers in another province in 2020, and rectal swabs from children in Ho Chi Minh City over three consecutive years, aiming to identify HVKP. Although it costs a lot of efforts, the sampling criteria based on different time and location lack scientific rationale, resulting in incomparable results.

As mentioned above, our primary objective was not to directly compare the prevalence of hvKp in bloodstream infections and healthy cohorts. Instead, our focus was on assessing the diversity and overlap of hvKp genotypes in BSIs and among healthy carriers. Such information is scarce in lowresource settings like Vietnam. Throughout the manuscript, we have avoided making explicit prevalence comparisons. In response to Reviewer 1's feedback on a similar concern, we have removed any sentences that may have implied such comparisons. Furthermore, we have made additional revisions in the Discussion section to temper any conclusions in this regard.

Furthermore, these data cannot represent the entire country of Vietnam, the title of this article needs to be reconsidered. Overall, I don't recommend this manuscript to be published in the journal for current stage. However, I list my comments below so the author could refer to them while revising the manuscript before re-submit it to other journals.

We appreciate the reviewer's consideration regarding the representativeness of our data and the title of our manuscript. As with other genomic surveillance studies, the issue of data representativeness is often debated. While we acknowledge that our study may not capture the entire spectrum of hvKp infections across Vietnam, the title accurately reflects the specific focus and findings within our defined context. Our study provides valuable insights into hvKp genotypes in the studied region, contributing to the existing literature on this topic. However, we will ensure to clarify the scope and limitations of our study in the manuscript to avoid any misinterpretation.

I agree it is necessary to screen for HVKP in healthy individuals and conduct such investigations in Vietnam. However, I do not believe that the study can adequately reflect the entire situation in Vietnam, and combining three entirely unrelated cohorts seems unnecessary. As you stated, 'We surmise that it is essential for infections and carriage data to originate from the same population and setting to draw meaningful conclusions.' If possible, I would propose undertaking a nationwide scientifically based sampling survey of healthy individuals. As for the current study, considering the limited number of HVKP cases from healthy individuals, it might be advisable to remove the healthy cohort and focus solely on the study of HVKP in bloodstream infections from one hospital in Vietnam over the past decade. This adjustment would help reduce unnecessary complexity and focus the research more effectively.

While our investigation of hvKp in bloodstream infections was conducted solely at a single hospital in Vietnam, it is crucial to highlight that this hospital serves as the largest referral center for tropical diseases in southern Vietnam. It caters not only to patients from Ho Chi Minh City but also to a significant population from neighboring provinces. Therefore, conducting research at this hospital offers a unique advantage, enabling us to capture the circulation of hvKp strains within a substantial catchment area.

We also wish to underscore the numerous challenges associated with conducting large-scale nationwide research involving healthy individuals in Vietnam, including the lack of baseline data and logistical complexities. We hypothesize that hvKp strains are disseminated throughout Vietnam and that the

population may have reached an equilibrium state. Thus, data from our selected population/cohorts likely provide insights into the extent of hvKp circulation in the general population. Furthermore, conducting surveillance spanning patients and healthy cohorts over a ten-year period would pose significant challenges.

We firmly believe that our surveillance data on healthy cohorts are relevant and offer valuable insights into dissecting the circulation of hvKp strains in the diseased population. Our findings can inform policymakers, public health officials, and clinical providers in devising appropriate strategies for diagnostics, treatment, and prevention measures to address this significant and often overlooked health threat.

1. To conduct a comparative study between children and adults, it would be more appropriate to classify and study the 700 cases of KP infection from the hospital according to age. As less HVKP was isolated from rectal swabs of healthy adults and children, the description in this regard can be considered for removal or downplaying.

Again, our primary focus in this study was not on KP infections or comparing children to adults. Rather, our research aimed to investigate hvKp and understand the diversity and overlap of hvKp genotypes in BSIs and among healthy carriers. It is important to note that while *K. pneumoniae* infections can occur in children (data not shown), all hvKp isolates from BSIs in our study were obtained from adults aged over 25 years old. The absence of hvKp infections in children is supported by the observation that no hvKp isolates were found in the rectal swabs obtained from healthy children. This finding aligns with existing literature, which has reported very few cases of hvKp infections in children. Our study findings contribute to understanding the reasons behind the low prevalence of hvKp infections occurring in children.

I can understand the desire to use of all the available data, but that doesn't mean we should simply lump everything together like a mishmash. If the goal is to report on the absence of HVKP in healthy children, why not write a separate dedicated article for that study?

The main objectives of our study were not centered on the absence of hvKp isolates in healthy children; rather, this outcome emerged as a result of our screening process within healthy cohorts. While acknowledging the significance of this finding, we argue against reporting solely on the absence of hvKp in healthy children in a separate article. Such reporting would lack the crucial disease-relevant context, particularly regarding the specific hvKp strains responsible for causing diseases. As explained above, the inclusion of healthy cohorts serves important purposes in our research, providing valuable insights into the circulation and epidemiology of hvKp strains.

2. It is recommended to download publicly available data from more public databases to enrich your phylogenetic tree analysis to include HVKP data from healthy individuals. Constructing a phylogenetic tree of various ST types together, as shown in Figure 1, makes little sense, because the results are likely to be clusters of the same ST type. A phylogenetic analysis focusing on the same ST would be more meaningful, as shown in Figure S3.

We thank the reviewer's suggestion regarding the inclusion of additional hvKp isolates from healthy individuals to strengthen our conclusions. However, it's important to note that there are currently no publicly available data from within Vietnam for such inclusion. We surmise that it is essential for infections and carriage data to originate from the same population and setting to draw meaningful conclusions.

The phylogenetic trees presented in Figure 1 and Figure S3 serve distinct purposes within our study. Figure 1 showcases the clustering of all sequence types (STs), providing an overview of their

distribution of virulence factors, acquired antimicrobial resistance (AMR) genes, and plasmid replicon types. This comprehensive view allows us to discern similarities and differences in key phenotypic and genotypic features both within and between STs. Additionally, the topology of this tree is compared with that of the plasmid tree to understand the potential for co-evolution and horizontal transfer of the virulence plasmid.

In contrast, the phylogenetic tree in Figure S3 specifically focuses on delineating the transmission dynamics of the ST23 population, a predominant clone circulating in Vietnam. Here, we highlight its transmission patterns, driven by multiple introductions and endemic expansion, providing insights into the spread of this particular lineage within the population.

Thanks for your explanation. Figure S3 seems to be more valuable than Figure 1, and I suggest moving Figure 1 to the supplementary files.

We acknowledge the value of Figure S3; however, we believe that Figure 1 holds crucial significance and effectively conveys the core findings of our study. Figure 1 provides a comprehensive overview of the clustering of all STs and their respective distribution of virulence factors, acquired AMR genes, and plasmid replicon types. Moreover, the tree topology is instrumental in comparing with the plasmid tree, elucidating the potential for co-evolution and horizontal transfer of the virulence plasmid among different STs, which is a key aspect of our investigation.

We intended to include Figure S3 in supplementary material to ensure that the focus of our audience remains on the main objectives of our study, particularly the exploration of genetic diversity and plasmid transfer mechanisms.

3. It is unclear from the description whether the phylogenetic analysis of strains differentiated between chromosomal and plasmid DNA. It is advised to focus on the chromosomal genome when analyzing the strains, excluding the interference from plasmids.

When analyzing the strains, we did focus exclusively on the chromosomal genome as outlined in the Methods section (line 228). We mapped our raw read data against the reference genome of hvKp strain NTUH -K2044 (accession number: AP006725), which contains solely the chromosomal genome. Furthermore, for the plasmid phylogeny analysis, we utilized only the virulence plasmid pLVPK (accession number: AY378100) as the reference genome (Line 262).

Thanks for clarification. Did you conduct data cleaning before mapping with the reads?

We performed quality check on the raw Illumina reads using FASTQC software, following our standard bioinformatic analysis pipeline. Additionally, adapters were removed prior to the mapping analysis.

4. I am unsure about the expression of plasmid types in the text. For instance, in line 299, what does "+/-" mean in "IncFIBK+/-IncHI1B"?

The virulence plasmid can exist as either a single replicon IncFIBK plasmid or a multi-replicon IncFIBK-IncHIB plasmid, as documented in previous publications (PMID: 33970792 and PMID: 30371343). We denote this as IncFIBK+/-IncHI1B, signifying that the IncFIBK replicon is consistently present on the virulence plasmid, while the presence of IncHI1B may vary among strains.

Thanks for clarification.

5. Could you please explain the reason for analysis of iuc3-carrying virulence plasmids? It is not the majority virulence plasmid in the study of HVKP in this article.

While iuc1 virulence plasmids are more prevalent in our study, iuc3 virulence plasmids exhibit epidemic potential due to their distinct genetic profile and self-transmissible nature. These iuc3

plasmids carry both virulence and conjugation modules, along with antimicrobial resistance genes, enabling them to spread in environments with high antibiotic selection pressure. Importantly, *iuc3*-carrying hvKp strains may not exhibit the hypermuroid phenotype, typically detected by traditional microbiology methods like the string test. This highlights the need for clinical microbiologists to be aware of alternative detection methods for *iuc3*-carrying strains. Furthermore, the majority of *iuc3* plasmids in hvKp are believed to originate from animal sources, suggesting a zoonotic reservoir that warrants further investigation on a broader scale, encompassing both hosts and transmission mechanisms.

Thank you for the explanation. When analyzing *iuc3*, it would be more convincing to find additional data in public databases to analyze together?

We have already analyzed our *iuc3* plasmids alongside those from publicly available databases. Initially, we retrieved genomic data for all *iuc3*-carrying hvKp isolates from Pathogenwatch and NCBI, totaling approximately 32 genomes. Subsequently, we narrowed down the selection to hvKp isolates harboring *iuc3*-encoding IncFIBK-IncFII plasmids for comparative analysis with our study isolates, as illustrated in Figure S4.

6. The result description in the article makes me confused. After describing the genetic structure of *iuc3* plasmid, the study performed experiments to investigate the *iuc1* plasmid. Is it possible to analyze the genetic structure and transfer mechanism of the *iuc1* plasmid in depth, instead of focusing on *iuc3* plasmid?

The genetic structure of *iuc1* virulence plasmids has been extensively studied (PMID: 15276215 and PMID: 29338592), whereas there have been limited reports on the genetic structures of *iuc3* virulence plasmids. We conducted experiments and described conjugation results for both *iuc1* and *iuc3* plasmids as outlined in the Results section (line 433). Given that *iuc3* plasmids are selftransmissible, the transfer mechanism was specifically examined for the non-conjugative *iuc1* plasmids.

Thank you for the explanation. Although I feel that the research on *iuc3* have little value, as the manuscript does not demonstrate an increasing trend of *iuc3* within hvkp.

We have provided the rationales for the analysis of *iuc3* virulence plasmid as above. This analysis highlights its epidemic potential and warrants attention from both the scientific and public health communities for closely monitoring the spread of *iuc3* carrying hvKp strains, such as those belonging to ST25. The low prevalence of *iuc3* positive hvKp strains may be attributed to factors such as the low sensitivity of detection methods or their limited circulation in human populations. However, it is noteworthy that the prevalence of *iuc3* positive strains may vary when considering other infectious disease syndromes besides bloodstream infections.

7. The results and legend of Figure 4 are somewhat difficult to comprehend. Why are there newly appeared plasmids (blue arrows)? Could it be contamination from other plasmid-carrying strains or changes in plasmid size after transfer? It requires re-experiment and further experimental validation or sequencing to provide clear explanations. Additionally, the strip positions of the green and white arrows are the same, but do they represent plasmids of different sizes?

We have revised the text and legend of Figure 4 to enhance clarity and facilitate data interpretation. The newly appearing blue arrows denote plasmids with sizes differing from those in the donor strains. Contamination is highly unlikely as the plasmid-carrying transconjugants must survive in selective media containing both tellurite and antibiotics. It is probable that these new plasmid bands are resulting from plasmid co-integration or recombination events, as demonstrated in previous

studies (PMID: 34294113, PMID: 35967858 and PMID: 34075192).

In each conjugation experiment, we randomly selected 5 colonies to compare the number and size of plasmid bands between donors, recipient, and transconjugants. It is evident that such plasmid cointegration or recombination events do not occur in all transfer events, a finding consistent with previous studies. Each conjugation experiment was conducted at least twice to confirm the transferability of the virulence plasmids, and the newly emerged bands were also observed in other sets of experiments (data not shown).

The white arrows correspond to a phagemid present in recipient and transconjugants, while the green arrows represent distinct phagemids found in the donors. Their strip positions appear similar because they are very similar in sizes (white=109,938 bp; green=109,376 bp/D2 strain, and 106,291 bp/D1 strain).

Thank you for the explanation. I suggest the plasmid profiling is best done with S1-PFGE and southern blotting.

We believe that our plasmid conjugation experiments were performed in accordance with a standard protocol and rigorous controls. The use of two selective markers with high concentrations of tellurite and ciprofloxacin/tetracycline for the selection of transconjugants has effectively minimized the risk of contamination. Additionally, our plasmid profiling data provide clear evidence of the transferability of the virulence plasmid, as demonstrated by its successful transfer between the donors and the recipient strain (Figure 4, specifically exemplified by donor 8-D8).

8. Please provide a summary of the highlights or significant discoveries in the abstract.

We have revised the abstract to highlight our significant discoveries.

While there have been articles documenting the significance of *nic* sites in plasmid transfer, it is essential to note that the sites identified in this study did not precisely match those previously reported. The distribution patterns of *nic* sites and plasmids observed in clinical strains do not definitively exclude the influence of other factors. To establish the functions of these sites, validation through mutants is necessary.

It is noteworthy that a recent study (PMID: 38042959) reported that *K. pneumoniae* carrying virulence genes may not necessarily exhibit high virulence, so defining hvKP should be more cautious.

For clarification, the majority of IncFIBK+/-IncHIB virulence plasmids in this study were found to harbor the common *nic* site "AGTTTGGTGC" of oriT type: OriT_100998, as previously reported in IncFIBK-IncHIB virulence plasmid (PMID: 34294113, Figure S6 & PMID: 35844192, Figure 3). There was only one strain carrying an IncFIBK-IncHIB virulence plasmid with a predicted novel *nic* site (oriT type: oriT_Q2), which also shared high sequence similarity in *nic* site of its helper plasmid (oriT_100015).

While there may be other mechanisms, the relevance and significance of *nic* sites in the transfer of non-conjugative virulence plasmid in hvKp have been showed in previous publications (PMID: 34294113; PMID: 35844192). In our study, we found 5 isolates carrying helper plasmids capable of aiding the transfer the virulence plasmid. Among them, three isolates harbored a helper plasmid with a nearly identical *nic* site (GGTGTGGTGA, oriT_100096) to that (GGTGTGGTGC) found in an incF helper plasmid previously demonstrated to facilitate the transfer of the virulence plasmid (PMID: 34294113). Therefore, it may be unnecessary to conduct a variety of mutation experiments for this purpose, as it could potentially complicate the manuscript and detract from its key message.

The paper you referred to (PMID: 38042959) assessed the virulence of hvKp strains using a murine pneumonia model, which is unrelated to our bloodstream infection study. It is important to note that the virulence factors contributing to pathogenesis in *K. pneumoniae* may differ between pneumonia and bloodstream infections (PMID: 31092506).